# Graph Inference Acceleration by Bridging GNNs and MLPs with Self-Supervised Learning

## Abstract

Graph Neural Networks (GNNs) have demonstrated their effectiveness in a variety of graph learning tasks such as node classification and link prediction. However, GNN inference mainly relies on neighborhood aggregation, which limits the deployment in latency-sensitive (i.e., real-time) applications such as financial fraud detection. To solve this problem, recent works have proposed to distill knowledge from teacher GNNs to student Multi-Layer Perceptrons (MLPs) trained on node content for inference acceleration. Despite the progress, these studies still suffer insufficient exploration of structural information when inferring unseen nodes. To address this issue, we propose a new method (namely **SSL-GM**) to fully integrate rich structural information into MLPs by bridging **G**NNs and **M**LPs with Self-Supervised Learning (**SSL**) for graph inference acceleration while improving model generalization capability. A key new insight of SSL-GM is that, without fetching their neighborhoods, the structural information of unseen nodes can be inferred solely from the nodes themselves with SSL. Specifically, SSL-GM employs self-supervised contrastive learning to align the representations encoded by graph context-aware GNNs and neighborhood dependency-free MLPs, fully integrating the structural information into MLPs. In particular, SSL-GM approximates the representations of GNNs using a non-parametric aggregator to avoid potential model collapse and exploits augmentation to facilitate the training; additionally, SSL-GM further incorporates reconstruction regulation to prevent representation shift caused by augmentation. Theoretically, we interpret our proposed SSL-GM through the principle of information bottleneck, demonstrating its generalization capability; we also analyze model capacity in incorporating structural information from the perspective of mutual information maximization and graph smoothness. Empirically, we demonstrate the superiority of SSL-GM over existing state-of-the-art models in both efficiency and effectiveness. In particular, SSL-GM obtains significant performance gains **(7~26%)** in comparison to MLPs, and a remarkable acceleration of GNNs **(90~126×)** on large-scale graph datasets.

## 1 Introduction

Due to the ubiquity of graph-structured data (e.g., computing networks, recommender systems in e-commerce, citation networks, and social networks), Graph Neural Networks (GNNs) have drawn significant attention in recent years. Generally, GNNs are based on neighborhood aggregation (Gilmer et al., 2017) to learn representations of given graphs. Despite their effectiveness in various graph learning tasks such as node classification and link prediction, due to the cost of neighborhood fetching during the testing stage, GNNs still face limitations of the deployment in latency-sensitive (i.e., real-time) applications such as financial fraud detection (Zhang et al., 2022; Wang et al., 2021). To solve this problem, existing works mainly adopt quantization (Ding et al., 2021), pruning (Zhou et al., 2021), and knowledge distillation (Yan et al., 2020) for graph inference acceleration. However, these improvements are limited as they still rely on neighborhood dependency (Zhang et al., 2022). To address this issue, as Multi-Layer Perceptrons (MLPs) have no dependency on graph data and can be efficiently deployed in latency-sensitive applications, researchers have explored distilling knowledge from pre-trained GNNs into MLPs (Zhang et al., 2022; Tian et al., 2023; Wang et al., 2023). Despite the progress, these methods inevitably sacrifice the model generalization capacity, as they cannot fully leverage structural information when inferring testing nodes. Given these challenges, we naturally ask: *how to bridge graph context-aware GNNs and neighborhood dependency-free MLPs for graph inference acceleration while improving model generalization capability?*

To answer the above question, we bring a key new insight different from existing works on graph inference acceleration: without fetching their neighborhoods, the structural information of unseen nodes can be inferred solely from nodes themselves with Self-Supervised Learning (SSL) (Chen et al., 2020b). Accordingly, we propose a new method (namely **SSL-GM**) to fully integrate rich structural information into MLPs by bridging **G**NNs and **M**LPs with Self-Supervised Learning (**SSL**) for graph inference acceleration while improving model generalization capability (Huang et al., 2023; Cabannes et al., 2023). More specifically, our proposed SSL-GM applies self-supervised contrastive learning (He et al., 2020) to align the consistency between GNNs and MLPs in the representation space. In particular, SSL-GM approximates the representations of GNNs using a non-parametric aggregator to avoid potential model collapse (Grill et al., 2020) and exploits augmentation to further enhance generalization (Zhao et al., 2021); additionally, SSL-GM further incorporates reconstruction regulation to prevent representation shift caused by augmentation. Theoretically, we have demonstrated that minimizing the objective function of SSL-GM is equivalent to optimizing the information bottleneck, thereby assuring model generalization (Alemi et al., 2017); in addition, we also analyze model capacity in incorporating structural information from the perspective of mutual information maximization and graph smoothness. Empirically, through extensive experiments over large-scale graph datasets, SSL-GM shows state-of-the-art performance on node classification tasks across transductive, inductive (Zhang et al., 2022), and cold-start settings. In terms of inference efficiency, SSL-GM exhibits remarkable acceleration compared to GNNs (**90∼126×**) and other acceleration techniques (**5∼90×**). The main contributions of our work are summarized below:

(**Methodology**) We observe that existing graph inference acceleration methods using low-latency MLPs are incapable of acquiring generalizable structure-aware representations. To this end, we propose SSL-GM to bridge graph context-aware GNNs and neighborhood dependency-free MLPs with SSL *at the first attempt* for graph inference acceleration while improving model generalization.

(**Theory**) We establish the theoretical equivalence between our objective and information bottleneck, proving the generalization capability of SSL-GM. Moreover, we analyze the capability of SSL-GM on encoding structural knowledge through mutual information maximization and graph smoothness.

(**Experiments**) Our SSL-GM achieves state-of-the-art performance over ten graph benchmarks on node classification tasks under transductive, inductive, and cold-start settings. It also shows a remarkable graph inference acceleration compared to GNNs (90∼126×) and exhibits significant performance improvements over vanilla MLPs (7∼26%).

## 2 RELATED WORK

**Graph Neural Networks** learn node representations by passing and aggregating messages from neighboring nodes. For example, GCN (Kipf & Welling, 2017) employs the normalized Laplacian matrix to guide message passing, GraphSAGE (Hamilton et al., 2017) utilizes neighborhood sampling, and GAT (Veličković et al., 2018) applies attention mechanisms. More recently, certain studies (Wu et al., 2019; Gasteiger et al., 2019; Han et al., 2023) decompose feature transformation from message passing, demonstrating that the effectiveness of GNNs stems from message propagation (Yang et al., 2023a). Despite their success, the inference speed on testing nodes remains a limitation due to neighborhood dependencies, which will be addressed in this paper.

**Self-Supervised Learning** (SSL) (Chen et al., 2020b; He et al., 2020) is a pre-training strategy that cultivates discriminative representations without supervision. Numerous studies (Veličković et al., 2019; Hassani & Khasahmadi, 2020; Zhu et al., 2020; 2021; Thakoor et al., 2022) have introduced methods for acquiring knowledge from graphs, prompting the downstream tasks (Sun et al., 2023). Particularly, BGRL (Thakoor et al., 2022) employs bootstrapping (Grill et al., 2020) to minimize the distance between node representations in two augmented views with an efficient approach. Despite potential improvements in generalization (Jiang et al., 2019; Huang et al., 2023), the dependency on neighborhood information continues to constrain inference speed.

**Inference Acceleration** encompasses quantization (Gupta et al., 2015; Jacob et al., 2018), pruning (Han et al., 2015; Frankle & Carbin, 2019), and knowledge distillation (KD) (Hinton et al., 2015). Quantization (Ding et al., 2021) approximates continuous data with limited discrete values, pruning (Zhou et al., 2021) involves removing connections within neural networks, and KD distills knowledge from large GNNs to small GNNs (Yan et al., 2020). However, these methods still fail to eliminate neighborhood dependencies, resulting in constrained inference acceleration. In light of this, GLNN (Zhang et al., 2022) distills knowledge from teacher GNNs to student MLPs, bypassing

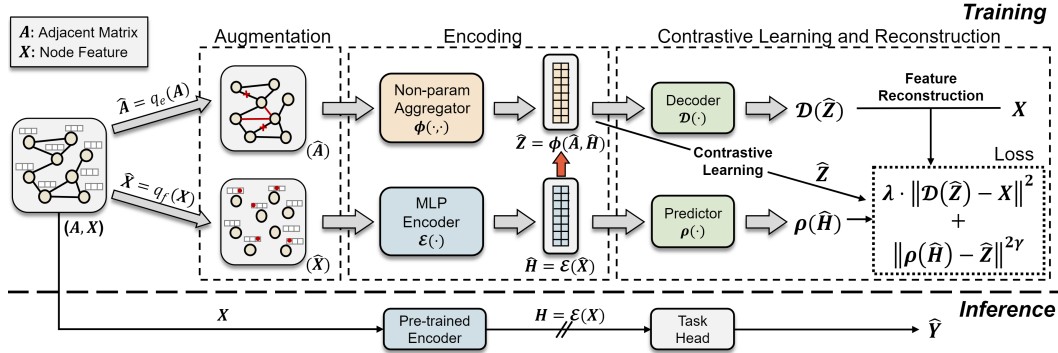

Figure 1: In training, SSL-GM augments the original graph and encodes representations through MLPs and GNNs, where the GNN representations are approximated by a non-parametric aggregator. Following, SSL-GM employs contrastive learning to maximize the alignment between these two representations and applies reconstruction based on GNN representations. In inference, only the MLP is leveraged to encode graph representations, leading to a substantial inference acceleration.

neighbor-fetching latency, but cannot fully model the structural information. Following studies integrate structural knowledge in pre-processing, e.g., appending positional embeddings (Perozzi et al., 2014) to node content (Tian et al., 2023; Wang et al., 2023), conducting label propagation (Yang et al., 2021), acquiring a motif cookbook (Yang et al., 2023b). However, these methods introduce additional time overhead and are impractical for out-of-distribution and cold-start generalization. In parallel, GraphMLP (Hu et al., 2021) and following works (Dong et al., 2022; Liu et al., 2022) employ neighborhood-aware distillation to train MLPs, albeit limited to the transductive setting. Unlike these methods, SSL-GM applies self-supervised learning to bridge GNNs and MLPs, achieving significant inference acceleration meanwhile enhancing model generalization.

## 3 BRIDGING GNNS AND MLPS WITH SELF-SUPERVISED LEARNING

In this section, we provide a detailed description of SSL-GM. Figure 1 illustrates an overview of the SSL-GM framework. The key insight is that, without fetching node neighborhoods, the potential structural distribution of unseen nodes can be inferred solely from node content with SSL. To achieve the goal, SSL-GM employs self-supervised contrastive learning to align the consistency between the representations encoded by GNNs and MLPs, integrating fine-grained structural information into MLPs to learn generalizable graph representations (Hendrycks et al., 2019; Cabannes et al., 2023).

**Problem Statement.** We consider a graph $\mathcal{G} = (\boldsymbol{A}, \boldsymbol{X})$ consisting of node set $\mathcal{V}$ and edge set $E$, with $N$ nodes in total. We have node features $\boldsymbol{X} \in \mathbb{R}^{N \times d}$ with dimension $d$, and adjacent matrix $\boldsymbol{A}$, where $\boldsymbol{A}_{ij} = 1$ if node $i$ and $j$ are connected, and $\boldsymbol{A}_{ij} = 0$ otherwise. Our focus lies in training an MLP encoder $\mathcal{E}(\cdot)$ without supervision, which receives node content $\boldsymbol{X}$ as input and generates node representations $\boldsymbol{H} \in \mathbb{R}^{N \times d'}$ preserving the semantics of both $\boldsymbol{A}$ and $\boldsymbol{X}$.

### 3.1 STRUCTURE-AWARE MLPS WITH SELF-SUPERVISED LEARNING

The effectiveness of GNNs stems from their ability to learn graph contextual information. Although some methods propose to distill knowledge from GNNs to MLPs, they inevitably sacrifice model capability (Tian et al., 2020) and generalization (Tian et al., 2023), as they only align the predictions of MLPs and GNNs in the label space, falling short to fully explore graph structural knowledge. To this end, we employ self-supervised contrastive learning to comprehensively incorporate structural information into MLPs by aligning GNNs and MLPs in the representation space. Specifically, we treat the representations encoded by MLPs as $\boldsymbol{H}$ and GNNs as $\boldsymbol{Z}$, which corresponds to the encodings on node view $\mathcal{G}_1 = (\emptyset, \boldsymbol{X})$ and graph view $\mathcal{G}_2 = (\boldsymbol{A}, \boldsymbol{X})$, respectively. The objective is to optimize the consistency between these two representations, thereby encoding structural knowledge into MLPs. We employ the Bootstrap loss (Grill et al., 2020) as objective function, defined as

$$\mathcal{L}_{cont} = \mathbb{E}\|\rho(\boldsymbol{H}) - \boldsymbol{Z}\|^{2\gamma}, \tag{1}$$

where $\gamma \geq 1$ serves as a scaling term, akin to an adaptive sample reweighing technique (Hou et al., 2022; Lin et al., 2017). The projector $\rho(\cdot)$ can either be identity or learnable. Here we opt for a non-linear MLP to enhance the expressiveness in estimating instance distances (Chen et al., 2020b).

**Non-Parametric Aggregator for Approximating GNN Representations.** Directly applying GNNs may lead to model collapse, as evidenced in Appendix D.3. We suppose it derives from the inconsistency between representations learned by MLPs and GNNs (He et al., 2020; Grill et al., 2020). To mitigate the issue, we propose to propagate the representations learned by MLPs to approximate the representations of GNNs, preserving their inherent consistency. Specifically, we employ a non-parametric aggregator $\phi(\cdot, \cdot)$ to conduct message passing as follows

$$\textbf{GNN}: \boldsymbol{Z} = GNN(\boldsymbol{A}, \boldsymbol{X}; \Theta) \Longrightarrow \textbf{SSL-GM}: \boldsymbol{Z} = \phi(\boldsymbol{A}, \boldsymbol{H}), \boldsymbol{H} = \mathcal{E}(\boldsymbol{X}; \Theta), \quad (2)$$

where $\Theta$ represents the parameters of the model. Unlike existing GNNs, our SSL-GM decomposes the linear transformation and message passing by applying MLP encoder to transform node features $\boldsymbol{H} = \mathcal{E}(\boldsymbol{X})$ and then employing the non-parametric aggregator to perform message passing $\boldsymbol{Z} = \phi(\boldsymbol{A}, \boldsymbol{H})$ without additional transformation. Further details are presented in Appendix D.6. This approach has been empirically (Wu et al., 2019; Gasteiger et al., 2019) and theoretically (Han et al., 2023; Yang et al., 2023a) shown be expressive. The choice of aggregation type can be arbitrary. We employ a GCN-like (Kipf & Welling, 2017) framework for neighborhood aggregation.

## 3.2 AUGMENTATION TO FACILITATE TRAINING

The fundamental assumption behind training MLPs on graphs is that nodes with similar features have similar surrounding ego-graphs (Chen et al., 2021). This aligns with our insight that the contextual neighborhoods can be inferred based on the target nodes themselves. However, the assumption indicates the necessity of high-quality node features (Zhang et al., 2022; Guo et al., 2023) and inherently requires the consistency in structural distribution between training and testing graphs (Luan et al., 2022; 2023). This prevents MLP-based graph learning algorithms from generalizing to out-of-distribution settings. To solve the issue, we augment the node and ego-graph pairs to increase the quality and diversity of training data (Feng et al., 2020; You et al., 2020; Zhao et al., 2021). This approach will enhance model generalization and robustness for graphs that originate from distributions different from the training set. The augmentation is applied in each training epoch as

$$\hat{\mathcal{G}} = (\hat{\boldsymbol{A}}, \hat{\boldsymbol{X}}), \hat{\boldsymbol{A}} \sim q_e(\boldsymbol{A}), \hat{\boldsymbol{X}} \sim q_f(\boldsymbol{X}), \quad (3)$$

where $q_e(\cdot)$ and $q_f(\cdot)$ are two random augmentation methods for graph structures and node features, respectively. This augmentation aligns with the objective of optimal contrastive learning (Xu et al., 2021) that aims to train an augmentation-invariant encoder, formulated as

$$\mathcal{E}^* = \arg\min_{\mathcal{E}} I(\mathcal{G}_1, \mathcal{G}_2) - I(\hat{\boldsymbol{H}}, \hat{\boldsymbol{Z}}), \quad (4)$$

where $\mathcal{G}_1 = (\emptyset, \hat{\boldsymbol{X}})$ and $\mathcal{G}_2 = (\hat{\boldsymbol{A}}, \hat{\boldsymbol{X}})$, with $\hat{\boldsymbol{H}}$ and $\hat{\boldsymbol{Z}}$ are representations encoded by MLPs and GNNs on augmented graphs, respectively. This process will facilitate the training of the MLP encoder $\mathcal{E}$ by incorporating more structure-relevant information into SSL-GM (Xu et al., 2021).

## 3.3 RECONSTRUCTION FOR MITIGATING REPRESENTATION SHIFT

While augmentation can facilitate the training process, it may impact the distribution of encoded representations. This representation shift, particularly pronounced in structural augmentation, can significantly alter the local structure of the target node. For instance, as depicted in Figure 2, simple edge permutation can dramatically change the 2-hop neighborhoods of a target node, leading to a substantial shift in the representations. Although this shift can potentially benefit existing graph contrastive learning methods by providing adversarial samples (Suresh et al., 2021; Kong et al., 2022; Feng et al., 2022), it may result in a mismatch between the augmented node and ego-graph pairs, thereby impairing the quality of representations learned by MLPs. To counter this problem, we hypothesize that if GNN representations can preserve localized information, the impact of representation shift can be minimized. Based on this, we introduce a reconstruction regularizer that reconstructs the raw node features $\boldsymbol{X}$ based on the GNN representations $\hat{\boldsymbol{Z}}$ on augmented graphs $\hat{\mathcal{G}} = (\hat{\boldsymbol{A}}, \hat{\boldsymbol{X}})$. The reconstruction term is defined as

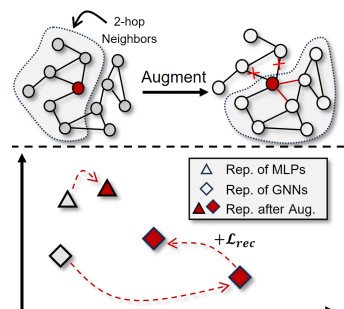

Figure 2: Augmentation leads to severe representation shift.

$$\mathcal{L}_{rec} = \mathbb{E}\|\mathcal{D}(\hat{\boldsymbol{Z}}) - \boldsymbol{X}\|^2, \hat{\boldsymbol{Z}} = \phi(\boldsymbol{A}, \hat{\boldsymbol{H}}), \hat{\boldsymbol{H}} = \mathcal{E}(\hat{\boldsymbol{X}}), \quad (5)$$

where $\mathcal{D}(\cdot)$ represents the decoder, while $\hat{\boldsymbol{Z}}$ and $\hat{\boldsymbol{H}}$ correspond to representations of GNNs and MLPs on the augmented graphs, respectively. This term encourages GNNs to preserve more localized information, ensuring GNN representations do not significantly shift from MLP representations, shown in Figure 2. However, when no augmentations are applied, the term may even lead to model performance degradation, evaluated in Appendix D.5. Additionally, the reconstruction regularizer also provides denoising capability (Batson & Royer, 2019) to some extent, enhancing the robustness of SSL-GM to noisy data.

## 3.4 OBJECTIVE FUNCTION

Considering the above three modules, we define the overall objective of SSL-GM as

$$\mathcal{L} = \mathcal{L}_{cont} + \lambda \cdot \mathcal{L}_{rec} = \mathbb{E}[\underbrace{\|\rho(\hat{\boldsymbol{H}}) - \hat{\boldsymbol{Z}}\|^{2\gamma}}_{invariance} + \lambda \cdot \underbrace{\|\mathcal{D}(\hat{\boldsymbol{Z}}) - \boldsymbol{X}\|^2}_{reconstruct}], \qquad (6)$$

where $\hat{\boldsymbol{H}} = \mathcal{E}(\hat{\boldsymbol{X}})$ and $\hat{\boldsymbol{Z}} = \phi(\hat{\boldsymbol{A}}, \hat{\boldsymbol{H}})$ are representations of MLPs and GNNs on the augmented graph $\hat{\mathcal{G}} = (\hat{\boldsymbol{A}}, \hat{\boldsymbol{X}})$. $\rho$ and $\mathcal{D}$ indicate projector head and decoder. $\gamma$ and $\lambda$ denote scaling factor and trade-off coefficient. We dubbed the contrastive loss as invariance term and the reconstruction loss as reconstruction term. While seemingly distinct, these two terms collaboratively reciprocate each other. The invariance term produces graph context-aware MLP representations, akin to positional encoding (Rampášek et al., 2022), improving the quality of GNN representations (Dwivedi & Bresson, 2020; Kreuzer et al., 2021; Dwivedi et al., 2022; Wang et al., 2022). This enhanced discrimination capability facilitates node feature reconstruction. The reconstruction term, in turn, prevents potential representation shifts by preserving additional localized information in representations learned by GNNs, thus providing better high-order structural signals for training MLPs.

**Theorem 1** *Suppose $\mathcal{G} = (\boldsymbol{A}, \boldsymbol{X})$ is sampled from a latent graph $\mathcal{G}_{\mathcal{I}} = (\boldsymbol{A}, \boldsymbol{F})$, $\mathcal{G} \sim P(\mathcal{G}_{\mathcal{I}})$ (Xie et al., 2022), and $\boldsymbol{F}^*$ is the lossless compression of $\boldsymbol{F}$ that $\mathbb{E}[\boldsymbol{X}|\boldsymbol{A}, \boldsymbol{F}^*] = \boldsymbol{F}$. Let $\mathcal{E}$ be a l-Lipschitz continuous function respect to $l_2$-norm, $\rho$ be an identity projector, and $\lambda = 1, \gamma = 1$. Optimizing Eq. 6 equals to finding the optimal compression $\boldsymbol{T}^*$ with minimal sufficient information $\boldsymbol{C}$ where*

$$\boldsymbol{T}^* = \arg\min_{\boldsymbol{T}} I(\mathcal{G}; \boldsymbol{T}) - \beta I(\boldsymbol{T}; \mathcal{G}_{\mathcal{I}}), s.t., I(\boldsymbol{T}, \mathcal{G}_{\mathcal{I}}) \geq \boldsymbol{C}, \boldsymbol{T} = (\boldsymbol{H}, \boldsymbol{Z}). \qquad (7)$$

Theorem 1 reveals the equivalence of optimization objectives between our loss function in Eq. 6 and information bottleneck (Tishby et al., 2000; Tishby & Zaslavsky, 2015), ensuring our SSL-GM to learn informative and generalizable representations (Alemi et al., 2017) for downstream tasks.

## 4 EXPERIMENTS

### 4.1 EXPERIMENTAL SETTINGS

**Datasets.** We evaluate our SSL-GM model on ten public benchmark datasets for node classification task, including Cora, Citeseer, Pubmed, Amazon-CS, Amazon-Photo, Coauthor-CS, Coauthor-Physics, Wiki-CS, Flickr, and Arxiv. Dataset details are presented in Appendix B.1. We also evaluate the model performance on graph classification datasets, shown in Appendix B.1 and D.1.

**Baselines.** We compare SSL-GM to a variety of baselines, including supervised GNNs, e.g., SAGE (Hamilton et al., 2017), GAT (Veličković et al., 2018), SGC (Wu et al., 2019), APPNP (Gasteiger et al., 2019), self-supervised methods, including DGI (Veličković et al., 2019), MVGRL (Hassani & Khasahmadi, 2020), GRACE (Zhu et al., 2020), BGRL (Thakoor et al., 2022), GCA (Zhu et al., 2021), and MLP-based methods, such as basic MLP, GraphMLP (Hu et al., 2021), GLNN (Zhang et al., 2022), GENN (Wang et al., 2023), and NOSMOG (Tian et al., 2023). Appendix B.2 presents the details.

**Evaluation Protocol.** We report the mean and standard deviation of ten runs using different random seeds to assess model performance in node classification. Accuracy serves as the metric. We conduct comparison for SSL-GM against baseline methods in transductive, inductive (production) (Zhang et al., 2022), and cold-start settings. Appendix B.3 and B.4 show additional information.

### 4.2 NODE CLASSIFICATION RESULTS

**Transductive Setting.** Table 1 demonstrates the node classification results of SSL-GM and baselines in transductive setting. Our SSL-GM consistently outperforms MLP-based methods across all datasets, which highlights the expressiveness of incorporating fine-grained structural knowledge

| | | Cora | Citeseer | PubMed | Amazon-CS | Amazon-Photo | Co-CS | Co-Phys | Wiki-CS | Flickr | Arxiv |
|---|---|---|---|---|---|---|---|---|---|---|---|
| GNN | SAGE | 81.44±0.91 | 70.44±1.39 | 85.94±0.43 | 88.88±0.27 | 93.81±0.42 | 93.41±0.16 | 95.72±0.05 | 80.87±0.63 | 48.45±0.78 | 72.05±0.25 |
| | GAT | 82.33±1.17 | 68.89±1.47 | 84.73±0.43 | 89.92±0.48 | 91.94±0.44 | 91.98±0.32 | 95.08±0.15 | 79.97±0.56 | 51.38±0.16 | 71.79±0.40 |
| | APPNP | 75.48±1.57 | 68.09±1.20 | 84.60±0.30 | 87.41±0.27 | 93.37±0.47 | 94.62±0.21 | 95.44±0.12 | 79.10±0.31 | 47.53±0.29 | 71.01±0.24 |
| | SGC | 81.80±0.88 | 68.96±1.63 | 85.28±0.32 | 89.31±0.62 | 92.74±0.43 | 94.02±0.20 | 94.78±0.20 | 81.06±0.55 | 51.75±0.24 | 69.95±0.35 |
| GCL | DGI | 82.31±0.60 | 71.81±0.73 | 76.78±0.70 | 79.98±0.19 | 91.60±0.21 | 92.22±0.53 | 94.50±0.04 | 76.42±0.55 | 46.88±0.13 | 70.13±0.15 |
| | MVGRL | 83.89±0.50 | 72.14±1.25 | 86.33±0.59 | 87.85±0.31 | 91.88±0.15 | 92.15±0.07 | 95.30±0.04 | 77.64±0.09 | 49.32±0.11 | 70.88±0.10 |
| | GRACE | 80.50±1.03 | 65.52±2.06 | 84.64±0.50 | 88.44±0.33 | 92.83±0.56 | 93.01±0.30 | 95.43±0.06 | 78.59±0.47 | 49.33±0.11 | 70.96±0.13 |
| | GCA | 83.53±0.49 | 71.33±0.15 | 86.03±0.37 | 87.42±0.30 | 92.61±0.21 | 93.06±0.03 | 95.72±0.05 | 78.35±0.05 | 49.03±0.07 | 70.90±0.08 |
| | BGRL | 81.30±0.59 | 66.90±0.58 | 84.92±0.24 | 88.19±0.21 | 92.54±0.11 | 92.11±0.12 | 95.21±0.07 | 77.54±0.79 | 49.67±0.06 | 70.84±0.12 |
| MLP | MLP | 64.49±1.90 | 64.01±1.26 | 80.69±0.28 | 80.79±0.33 | 87.77±0.49 | 91.65±0.32 | 95.11±0.12 | 75.16±0.46 | 46.21±0.07 | 56.44±0.30 |
| | GraphMLP | 79.50±0.81 | 72.10±0.48 | 84.27±0.23 | 84.01±0.58 | 90.90±1.03 | 90.36±0.64 | 93.51±0.15 | 76.39±0.53 | 46.25±0.21 | 63.36±0.18 |
| | GLNN | 81.32±1.15 | 71.15±0.71 | 86.34±0.46 | 87.47±0.60 | 93.87±0.31 | 93.51±0.05 | 95.40±0.07 | 80.66±0.74 | 46.18±0.19 | 64.03±0.51 |
| | GENN | 82.13±0.77 | 71.42±1.31 | 86.28±0.31 | 87.12±0.55 | 93.64±0.65 | 93.82±0.29 | 95.45±0.05 | 80.48±0.74 | 46.35±0.34 | 70.13±0.60 |
| | NOSMOG | 82.27±1.13 | 72.39±1.27 | 86.18±0.33 | 87.64±1.14 | 93.94±0.47 | 93.83±0.23 | 95.74±0.12 | 80.53±0.77 | 46.69±0.25 | 70.84±0.44 |
| | **SSL-GM** | **84.60±0.24** | **73.52±0.53** | **86.99±0.09** | **88.46±0.16** | **94.28±0.06** | **94.87±0.07** | **96.17±0.03** | **81.21±0.13** | **49.85±0.09** | **71.12±0.10** |
| | $\Delta_{\text{BGRL}}$ | ↑ **4.06%** | ↑ **9.90%** | ↑ 2.44% | ↑ 0.31% | ↑ 1.88% | ↑ 3.00% | ↑ 1.01% | ↑ **4.73%** | ↑ 0.36% | ↑ 0.40% |
| | $\Delta_{\text{MLP}}$ | ↑ **31.18%** | ↑ **14.86%** | ↑ **7.81%** | ↑ **9.49%** | ↑ **7.42%** | ↑ 3.51% | ↑ 1.11% | ↑ **8.05%** | ↑ **7.88%** | ↑ **26.01%** |
| | $\Delta_{\text{NOSMOG}}$ | ↑ 2.83% | ↑ 1.56% | ↑ 0.94% | ↑ 0.94% | ↑ 0.36% | ↑ 1.11% | ↑ 0.45% | ↑ 0.84% | ↑ **6.77%** | ↑ 0.40% |
| | w/o Aggr. | 55.91±0.66 | 57.36±0.33 | 79.93±0.32 | 72.76±0.71 | 77.05±0.18 | 91.19±0.13 | 93.35±0.12 | 73.87±0.26 | 45.82±0.07 | 54.83±0.41 |
| | w/o Pred. | 81.78±0.30 | 73.09±0.25 | 85.33±0.10 | 83.12±0.25 | 91.25±0.27 | 93.32±0.08 | 94.98±0.06 | 76.13±0.22 | 48.31±0.14 | 67.48±0.44 |
| | w/o Aug. | 82.10±0.45 | 71.83±0.43 | 86.89±0.13 | 87.12±0.15 | 93.52±0.20 | 93.10±0.05 | 94.56±0.06 | 80.98±0.13 | 48.21±0.10 | 70.58±0.20 |
| | w/o Rec. | 84.37±0.27 | 73.18±0.24 | 86.86±0.10 | 88.25±0.07 | 94.15±0.07 | 94.64±0.06 | 96.01±0.07 | 81.10±0.13 | 49.60±0.11 | 70.38±0.22 |

Table 1: Node classification accuracy (%) under transductive setting. $\Delta_{\text{BGRL}}$, $\Delta_{\text{MLP}}$, and $\Delta_{\text{NOSMOG}}$ represent the performance gap (%) between our methods and MLP, BGRL, and NOSMOG, where green indicates the improvement over 4%.

| | Cora | Citeseer | PubMed | Amazon-CS | Amazon-Photo | Co-CS | Co-Phys | Wiki-CS | Flickr | Arxiv |
|---|---|---|---|---|---|---|---|---|---|---|
| SAGE | 77.51±1.77 | 68.40±1.61 | 85.04±0.44 | 87.24±0.43 | 93.20±0.45 | 92.88±0.40 | 95.74±0.12 | 79.26±0.65 | 47.17±0.73 | 68.52±0.56 |
| BGRL | 77.73±1.07 | 64.33±1.56 | 83.97±0.48 | 87.33±0.48 | 91.47±0.62 | 91.26±0.35 | 94.38±0.29 | 76.25±1.09 | 49.12±0.31 | 69.29±0.38 |
| MLP | 63.76±1.65 | 63.98±1.22 | 80.91±0.45 | 81.00±0.54 | 87.73±0.88 | 91.68±0.59 | 95.18±0.13 | 75.08±0.71 | 46.14±0.22 | 55.89±0.51 |
| GLNN | 78.34±1.04 | 69.61±1.13 | 85.44±0.48 | 87.04±0.50 | 93.28±0.43 | 93.72±0.35 | 95.76±0.09 | 78.39±0.54 | 46.11±0.07 | 63.53±0.48 |
| GENN | 77.83±1.57 | 67.30±1.48 | 84.34±0.47 | 85.75±1.20 | 92.09±0.96 | 93.57±0.37 | 95.67±0.06 | 78.27±1.01 | 45.56±0.51 | 68.52±0.54 |
| NOSMOG | 77.83±1.94 | 68.58±1.41 | 83.84±0.45 | 86.61±1.22 | 92.52±0.68 | 93.45±0.44 | 95.78±0.10 | 78.35±0.70 | 46.05±0.55 | 69.10±0.80 |
| **SSL-GM** | **81.37±1.20** | **72.33±0.90** | **86.47±0.28** | **87.65±0.40** | **93.87±0.32** | **94.63±0.16** | **96.04±0.12** | **79.26±0.83** | **49.27±0.18** | **70.23±0.47** |
| $\Delta_{\text{BGRL}}$ | ↑ **4.68%** | ↑ **12.44%** | ↑ 2.98% | ↑ 0.37% | ↑ 2.62% | ↑ 3.69% | ↑ 1.76% | ↑ 3.95% | ↑ 0.31% | ↑ 1.36% |
| $\Delta_{\text{MLP}}$ | ↑ **27.62%** | ↑ **13.05%** | ↑ **6.87%** | ↑ **8.21%** | ↑ **7.00%** | ↑ 3.22% | ↑ 0.90% | ↑ **5.57%** | ↑ **6.78%** | ↑ **25.66%** |
| $\Delta_{\text{NOSMOG}}$ | ↑ **4.55%** | ↑ **5.47%** | ↑ 3.14% | ↑ 1.20% | ↑ 1.46% | ↑ 1.26% | ↑ 0.27% | ↑ 1.16% | ↑ **6.99%** | ↑ 1.64% |

Table 2: Node classification accuracy (%) under inductive (production) settings.

| | Cora | Citeseer | PubMed | Amazon-CS | Amazon-Photo | Co-CS | Co-Phys | Wiki-CS | Flickr | Arxiv |
|---|---|---|---|---|---|---|---|---|---|---|
| SAGE | 60.23±5.03 | 56.62±5.10 | 77.98±1.53 | 61.01±4.51 | 59.52±8.02 | 91.30±0.84 | 94.64±0.88 | 52.73±7.93 | 41.06±2.25 | 43.47±2.53 |
| BGRL | 78.80±1.14 | 65.10±2.08 | 84.18±0.80 | 86.13±0.76 | 90.39±0.30 | 90.23±0.48 | 94.06±0.28 | 78.15±1.17 | 48.73±0.13 | 64.11±0.20 |
| MLP | 64.15±2.11 | 64.43±1.76 | 80.90±0.72 | 80.80±0.91 | 87.88±0.96 | 91.78±0.81 | 95.16±0.18 | 74.94±1.81 | 46.09±0.50 | 55.91±0.69 |
| GLNN | 71.96±1.68 | 69.14±2.58 | 84.42±0.87 | 83.98±0.70 | 91.05±0.49 | 93.34±0.47 | 95.70±0.09 | 77.64±1.42 | 46.05±0.43 | 60.55±0.55 |
| GENN | 69.06±4.80 | 65.44±2.33 | 78.19±2.08 | 79.44±1.66 | 90.18±0.62 | 93.54±0.55 | 95.55±0.25 | 67.31±1.66 | 45.24±0.72 | 61.30±0.59 |
| NOSMOG | 70.69±2.45 | 68.03±2.79 | 81.48±1.30 | 81.95±1.04 | 91.15±0.88 | 93.63±0.42 | 95.54±0.40 | 68.49±3.61 | 46.07±0.30 | 61.64±0.93 |
| **SSL-GM** | **80.48±2.15** | **72.81±1.61** | **86.44±0.51** | **87.58±0.99** | **93.91±0.58** | **94.51±0.15** | **95.97±0.24** | **78.46±1.48** | **49.41±0.46** | **66.13±1.05** |
| $\Delta_{\text{BGRL}}$ | ↑ 2.13% | ↑ **11.84%** | ↑ 2.68% | ↑ 1.68% | ↑ 3.89% | ↑ **4.74%** | ↑ 2.03% | ↑ 0.40% | ↑ 1.40% | ↑ 3.15% |
| $\Delta_{\text{MLP}}$ | ↑ **25.46%** | ↑ **13.01%** | ↑ **6.85%** | ↑ **8.39%** | ↑ **6.86%** | ↑ 2.97% | ↑ 0.85% | ↑ **4.70%** | ↑ **7.20%** | ↑ **18.28%** |
| $\Delta_{\text{NOSMOG}}$ | ↑ **13.85%** | ↑ **7.03%** | ↑ **6.09%** | ↑ **6.87%** | ↑ 3.03% | ↑ 0.94% | ↑ 0.45% | ↑ **14.56%** | ↑ **7.25%** | ↑ **7.28%** |

Table 3: Node classification accuracy (%) under cold-start setting.

into MLPs. More specifically, SSL-GM surpasses MLP, GraphMLP, and GLNN on large-scale dataset Arxiv by 26%, 12%, and 11% improvements. Furthermore, SSL-GM outperforms other self-supervised methods in all datasets, and fully-supervised GNNs in 7 out of 10 datasets. In particular, SSL-GM achieves superior performance compared to SGC and APPNP, both of which employ propagators to encode node representations in a manner similar to our aggregator. To further analyze the expressiveness of SSL-GM, we conduct graph classification in Appendix D.1. Experimental results show our SSL-GM achieves the best or sub-best performance across 6 out of 7 datasets.

**Inductive (Production) Setting.** Table 2 presents the node classification results of SSL-GM and baseline methods in inductive (production) setting. We partition the original graph into two non-overlapping sets, namely the transductive set $\mathcal{G}^T = (\mathcal{V}^T, E^T)$ and the inductive set $\mathcal{G}^I = (\mathcal{V}^I, E^I)$, each with distinct structural distributions. We evaluate transductive and inductive results on these sets and then interpolate them to derive the production results. Unlike Tian et al. (2023), which establish connections between nodes in $\mathcal{V}^I$ and $\mathcal{V}^T$ during inductive inference, our setting is more challenging as it treats these two sets as independent from each other, simulating an *out-of-distribution* scenario. Appendix B.4 includes more details. We report the production results in Table 2 and provide the comprehensive results in Appendix D.2. SSL-GM outperforms all baselines in all datasets, demonstrating the effectiveness of SSL-GM in real-world settings. Surprisingly, GLNN outper-

forms the advanced NOSMOG on 8 out of 10 datasets. We suppose that the reliance on additional positional embeddings hinders the generalization of NOSMOG on graphs with distinct structures.

**Cold-start Setting.** In latency-constrained systems, newly emerged nodes may become isolated (Hao et al., 2021; Zheng et al., 2022), which is commonly referred to as the cold-start issue. Our SSL-GM, which infers potential structural information of newly emerged nodes solely based on node content, provides a promising solution to address this issue. To simulate the cold-start scenario, we follow the inductive (production) setting while removing connections within the inductive set. This ensures that all inductive nodes remain isolated, forming a cold-start set $\mathcal{G}^C = (\mathcal{V}^I, \emptyset)$. Table 3 presents the cold-start performance results for SSL-GM and baselines. SSL-GM achieves notable improvements compared to all baselines. Specifically, SSL-GM demonstrates performance improvements of 7% and 18% over MLP on the Flickr and Arxiv datasets and achieves 7% and 7% enhancements over NOSMOG. We suppose the deficiency of NOSMOG derives from the absence of positional embedding. Note that BGRL, which employs augmentation in model training, also achieves exceptional performance, even if it highly relies on structural knowledge.

## 4.3 INFERENCE ACCELERATION

Our primary aim of SSL-GM is to accelerate inference. In this section, we demonstrate the capability of SSL-GM by illustrating the trade-off between prediction accuracy and model inference time using the Arxiv dataset under a cold-start setting, as depicted in Figure 3. We observe that SSL-GM attains the best trade-off between accuracy and inference time. Compared to baselines with similar inference times, SSL-GM outperforms them significantly, achieving an accuracy of 66%, whereas NOSMOG and MLPs only reach 62% and 56% accuracy, respectively. In contrast, methods that achieve performance similar to SSL-GM demand a substantial amount of inference time. For example, the 2-layer BGRL (BGRL-L2) requires 314.7ms, and the 3-layer BGRL (BGRL-L3) requires 635.9ms, while SSL-GM only needs 2.5ms, resulting in an acceleration of $125\times$ and $254\times$, respectively. In the case of NOSMOG, increasing the hidden dimension

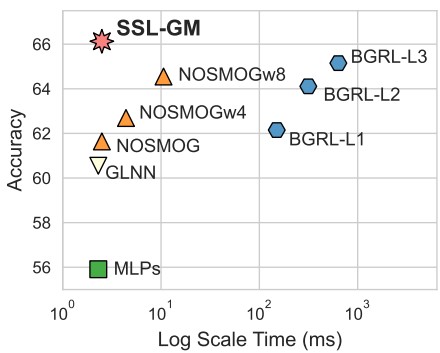

Figure 3: Accuracy vs. Inference Time on Arxiv dataset under cold-start setting.

(Zhang et al., 2022) can enhance performance by preserving more task-relevant information. We conduct a comparison between SSL-GM and NOSMOGw4 (four times wider than NOSMOG) and NOSMOGw8 (eight times wider than NOSMOG). We find that the wider models perform even worse than SSL-GM and demand more inference time. Consequently, we conclude that our SSL-GM surpasses existing baselines in accuracy while maintaining competitive inference speed. Additionally, we demonstrate that SSL-GM surpasses other acceleration methods in inference time on Flickr and Arxiv datasets ($5\sim90\times$) in Appendix C.

## 4.4 ABLATION STUDY

**Model Component.** In this section, we examine the impact of each component in the model. We provide a detailed analysis in Appendix D.3, D.4, D.5, and D.6. Table 1 shows the ablation results for ten benchmark datasets. The abbreviations `Aggr.`, `Pred.`, `Aug.`, and `Rec.` stand for aggregator, predictor, augmentation, and reconstruction. We observe the aggregator significantly enhances the model performance by injecting fine-grained structural information into the MLP encoder. The projector, which facilitates distance measurements between node representations, indeed improves the model performance, consistent with the findings in (Chen et al., 2020b). The augmentations improve the training of SSL-GM by synthesizing diverse node and ego-graph pairs during training. Moreover, the reconstruction term improves model performance by preventing the representation shifts of GNNs. Additionally, we implement the supervised version of SSL-GM in Appendix E.

**Robustness of SSL-GM to Noisy Data and Label Sparsity.** The quality of MLP encoder depends on both graph structure and node content. In this section, we introduce noise into the original graph to assess the robustness of SSL-GM on noisy data. Furthermore, we analyze the model performance

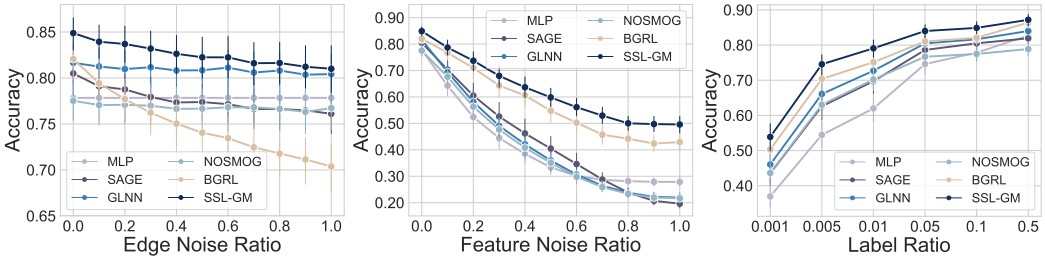

Figure 4: **Left**: Edge Noise. SSL-GM consistently demonstrates robustness against edge noise, even in scenarios with exceptionally high noise ratios. **Middle**: Feature Noise. SSL-GM demonstrates robustness to feature noise, whereas other MLP-based methods are susceptible to it. **Right**: Label Sparsity. SSL-GM significantly outperforms other baselines with a low label ratio.

with sparse labels. We report results on inductive set of production setting, averaged over seven datasets, including Cora, Citeseer, PubMed, Amazon-CS, Amazon-Photo, Co-CS, and Wiki-CS.

**(Noisy Topology)** To introduce structural noise, we randomly flip edges within the graph. Specifically, we replace $\boldsymbol{A}$ with $\tilde{\boldsymbol{A}} = \boldsymbol{M} \odot (1 - \boldsymbol{A}) + (1 - \boldsymbol{M}) \odot \boldsymbol{A}$, $M_{ij} \sim \mathcal{B}(p)$, where $\mathcal{B}(p)$ is a Bernoulli distribution with probability $p$. The results under various noise levels are depicted in Figure 4 (**Left**). Our SSL-GM consistently outperforms others, demonstrating its robustness. The increase of noise level leads to substantial performance degradation for GNNs, particularly for self-supervised BGRL, but imposes minimal impact on MLPs, even $\tilde{\boldsymbol{A}}$ becomes independent to $\boldsymbol{A}$.

**(Noisy Node Features)** Following, we examine the impact of node feature noise by introducing random Gaussian noise. We replace $\boldsymbol{X}$ with $\tilde{\boldsymbol{X}} = (1 - \alpha)\boldsymbol{X} + \alpha x$, where $x$ represents random noise agnostic to $\boldsymbol{X}$, and noise level $\alpha \in [0, 1]$. As depicted in Figure 4 (**Middle**), SSL-GM surpasses all baselines in all settings, even though node content quality is a crucial factor for MLP-based methods (Zhang et al., 2022; Guo et al., 2023). We attribute this robustness to the augmentation that synthesizes additional high-quality node and ego-graph pairs, thereby aiding MLP training. This augmentation also contributes to the robustness of BGRL. However, we observe that the performance of other MLP-based methods degrades rapidly with the increase in noise levels.

**(Label Sparsity)** In this section, we examine the robustness of SSL-GM under label sparsity. Figure 4 (**Right**) shows the model performance under various label ratios for node classification. We observe our method consistently outperforms all other baselines, even with extremely limited training data (0.001). This highlights the robustness of SSL-GM to label sparsity. Furthermore, we also observe that self-supervised methods exhibit better robustness Huang et al. (2023) compared to supervised methods, raised from the ability to leverage unlabeled data during training.

## 5   HOW SSL-GM LEARN FROM STRUCTURAL KNOWLEDGE?

**Mutual Information Maximization.** We interpret the SSL-GM from the perspective of information theory to theoretically analyze how MLPs encode structural information. Given a graph $\mathcal{G} = (\boldsymbol{X}, \boldsymbol{A}, \boldsymbol{Y})$ with node features $\boldsymbol{X}$, graph structure $\boldsymbol{A}$ and labels $\boldsymbol{Y}$, single MLP aims to minimize the cross-entropy between $\boldsymbol{X}$ and $\boldsymbol{Y}$, which corresponds to maximize the mutual information $\sum_{i \in \mathcal{V}} I(\boldsymbol{y}_i; \boldsymbol{x}_i)$ (Boudiaf et al., 2020) while disregarding the impact of structure. GNNs follows message passing framework that leverages subgraphs surrounding the target nodes to make prediction. We define subgraph around node $i$ as $\mathcal{S}_i = (\boldsymbol{X}^{[i]}, \boldsymbol{A}^{[i]})$, where $\boldsymbol{X}^{[i]}$ is the node features for neighborhoods of node $i$, and $\boldsymbol{A}^{[i]}$ is the adjacent matrix that describes the aggregation rule. Thus, optimizing GNNs equals to maximize $\sum_{i \in \mathcal{V}} I(\boldsymbol{y}_i; \mathcal{S}_i) = \sum_{i \in \mathcal{V}} I(\boldsymbol{y}_i; \boldsymbol{X}^{[i]}) + \sum_{i \in \mathcal{V}} I(\boldsymbol{y}_i; \boldsymbol{A}^{[i]} | \boldsymbol{X}^{[i]})$, which models the correlation between label $\boldsymbol{y}$ and both node feature $\boldsymbol{X}$ and graph structure $\boldsymbol{A}$. The objective of GLNN is to maximize $\sum_{i \in \mathcal{V}} I(\boldsymbol{x}_i; \boldsymbol{y}_i | \mathcal{S}_i)$, where $\boldsymbol{y}_i | \mathcal{S}_i$ denotes the soft labels given by GNNs. However, the approach cannot directly model the correlation between subgraph $\mathcal{S}$ and label $\boldsymbol{y}$, preventing to acquire structural knowledge. Some models, e.g., GENN and NOSMOG, utilize positional encoding to incorporate structural knowledge based on GLNN, whose objective is to maximize $\sum_{i \in \mathcal{V}} I(\boldsymbol{x}_i; \boldsymbol{y}_i | \mathcal{S}_i) + I(\boldsymbol{y}_i; \boldsymbol{A}^{[i]})$. Although they capture the

structural knowledge to some extent, they only model the correlation between label $\boldsymbol{y}$ and the graph structure $\boldsymbol{A}$ instead of the subgraph $\mathcal{S}$, failing to model fine-grained structural knowledge.

Unlike these models, the aim of our SSL-GM is to maximize $\sum_{i \in \mathcal{V}} I(\boldsymbol{y}_i; \boldsymbol{x}_i | \mathcal{S}_i) + I(\boldsymbol{x}_i; \mathcal{S}_i)$. The first term optimizes the model on downstream tasks and the second term is the objective of SSL-GM. We argue that when the second term is maximized, the objective technically equals to maximize $\sum_{i \in \mathcal{V}} I(\boldsymbol{y}_i; \mathcal{S}_i)$, which corresponds to the objective of GNNs. The objective ensures SSL-GM comprehensively leverages graph information $\mathcal{S}$, including node content and graph structure, in downstream tasks. Our analysis also aligns with the findings in Chen et al. (2021) and Zhang et al. (2022) that the expressiveness of GNNs and MLPs are theoretically bounded by the equivalence classes of induced rooted graphs, which corresponds to $\mathcal{S}$ in our case. Furthermore, SSL-GM augments node and ego-graph pairs to enhance the diversity of the training data, which inherently improves the quality of $\mathcal{S}$ and thereby improve the optimizing process of our SSL-GM.

**Graph Smoothness.** In addition to theoretical analysis, we empirically demonstrate the capability of SSL-GM in learning structural information as an inductive bias, which can be measured by graph smoothness. A low smoothness value indicates that representations of closely connected nodes are similar, allowing the model to extract more information from the graph data (Hou et al., 2019). We employ the Mean Average Distance (MAD) $\mathcal{L}_{MAD} = $

|         | Cora   | Citeseer | PubMed | Amazon-CS | Amazon-Photo | Average |
|---------|--------|----------|--------|-----------|--------------|---------|
| Raw Feat. | 0.8221 | 0.7825 | 0.7342 | 0.5393 | 0.5399 | 0.6836 |
| SAGE    | 0.1132 | 0.1835 | 0.1426 | 0.1564 | 0.1089 | 0.1409 |
| BGRL    | 0.1553 | 0.1023 | 0.3326 | 0.2509 | 0.2031 | 0.2088 |
| MLP     | 0.4633 | 0.4442 | 0.4853 | 0.4557 | 0.4317 | 0.4560 |
| GLNN    | 0.2818 | 0.2684 | 0.4208 | 0.3549 | 0.3976 | 0.3447 |
| NOSMOG  | 0.2672 | 0.2301 | 0.3942 | 0.3056 | **0.2773** | 0.2949 |
| **SSL-GM** | **0.1964** | **0.1703** | **0.3604** | **0.2986** | 0.2878 | **0.2627** |

Table 4: The graph smoothness value measures the representation similarity between nodes and their neighborhoods. Lower values indicate a smoother graph, signifying that the model can capture more structural knowledge.

$\frac{\sum_{v \in \mathcal{V}} \sum_{j \in \mathcal{N}(i)} (\boldsymbol{H}_i - \boldsymbol{H}_j)^2}{\sum_{v \in \mathcal{V}} \sum_{j \in \mathcal{N}(i)} \mathbf{1}}$ (Chen et al., 2020a) to quantify graph smoothness. For supervised methods, the representations are extracted from the output of the final layer before the prediction head. Table 4 presents the graph smoothness values for SSL-GM and the baseline methods. We observe that message passing methods inherently yield lower smoothness values than MLP-based methods. SSL-GM, which pulls close the outputs of MLPs and GNNs in the representation space, achieves an average smoothness value of 0.2627. This value is lower than that of MLP (0.4560), GLNN (0.3447), and NOSMOG (0.2949), indicating the superiority of learning fine-grained structural information. Note that a low smoothness value might potentially lead to over-smoothing (Li et al., 2019), but this topic falls outside the scope of this paper. We utilize smoothness to indicate whether the encoded representations can accurately reflect the graph structure.

**Normalized Cut.** Graph smoothness evaluates the consistency between representations and graph structure. To further analyze the alignment between predictions and structure, we adopt normalized cut, which approximates the min-cut problem. This problem involves minimizing the number of removed edges to partition nodes $\mathcal{V}$ into $K$ disjoint subsets. The min-cut problem can be formulated

|         | Cora   | Citeseer | PubMed | Amazon-CS | Amazon-Photo | Average |
|---------|--------|----------|--------|-----------|--------------|---------|
| SAGE    | 0.9243 | 0.9426 | 0.9177 | 0.8541 | 0.8721 | 0.9022 |
| BGRL    | 0.8847 | 0.9346 | 0.8556 | 0.8336 | 0.8493 | 0.8716 |
| MLP     | 0.6663 | 0.8035 | 0.8625 | 0.7183 | 0.7467 | 0.7595 |
| GLNN    | 0.8863 | 0.9162 | 0.7934 | 0.8038 | 0.8113 | 0.8422 |
| NOSMOG  | 0.9023 | 0.9317 | 0.8337 | 0.8384 | 0.8226 | 0.8657 |
| **SSL-GM** | **0.9335** | **0.9575** | **0.8863** | **0.9014** | **0.8604** | **0.9078** |

Table 5: Our model predictions are more consistent with the graph structure in terms of normalized cut value, although the model is trained in an unsupervised manner.

as $\mathcal{L}_{cut} = \frac{tr(\hat{\boldsymbol{Y}}^T \boldsymbol{A} \hat{\boldsymbol{Y}})}{tr(\hat{\boldsymbol{Y}}^T \boldsymbol{D} \hat{\boldsymbol{Y}})}$ where $\hat{\boldsymbol{Y}}$ represents the model predictions. A higher $\mathcal{L}_{cut}$ value indicates a better alignment between model predictions and the underlying structure. Table 5 presents the evaluation of min-cut scores for SSL-GM and baselines across five datasets. We observe message passing methods generally outperform MLPs due to their explicit encoding of structure. Remarkably, SSL-GM surpasses existing MLP-based methods and is even competitive with message passing models, demonstrating its superior ability to capture structural information.

## 6 CONCLUSION

We analyze that existing MLP-based graph inference acceleration methods cannot learn generalizable structure-aware representations, and propose a novel framework SSL-GM to solve it. The key insight is that the potential structural information of unseen nodes can be inferred solely based on the node content with SSL. Specifically, we employ self-supervised contrastive learning to align GNNs and MLPs in the representation space to integrate rich structural information into MLPs. Additionally, we apply non-parametric aggregator, graph augmentation, and feature reconstruction to improve model performance. We also empirically and theoretically demonstrate the expressiveness, generalization, and robustness of SSL-GM. In the future, we will extend SSL-GM to real-world applications, such as financial fraud detection. We hope our work can inspire researchers seeking to develop novel learning algorithms for graphs that go beyond the scope of GNNs.

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

## A  PROOF OF THEOREM 1

**Theorem 1** *Suppose $\mathcal{G} = (\boldsymbol{A}, \boldsymbol{X})$ is sampled from a latent graph $\mathcal{G}_{\mathcal{I}} = (\boldsymbol{A}, \boldsymbol{F})$, $\mathcal{G} \sim P(\mathcal{G}_{\mathcal{I}})$ (Xie et al., 2022), and $\boldsymbol{F}^*$ is the lossless compression of $\boldsymbol{F}$ that $\mathbb{E}[\boldsymbol{X}|\boldsymbol{A}, \boldsymbol{F}^*] = \boldsymbol{F}$. Let $\mathcal{E}$ be a l-Lipschitz continuous function respect to $l_2$-norm, $\rho$ be an identity projector, and $\lambda = 1, \gamma = 1$. Optimizing Eq. 6 equals to finding the optimal compression $\boldsymbol{T}^*$ with minimal sufficient information $\boldsymbol{C}$ where*

$$\boldsymbol{T}^* = \arg\min_{\boldsymbol{T}} I(\mathcal{G}; \boldsymbol{T}) - \beta I(\boldsymbol{T}; \mathcal{G}_{\mathcal{I}}), s.t., I(\boldsymbol{T}, \mathcal{G}_{\mathcal{I}}) \geq \boldsymbol{C}, \boldsymbol{T} = (\boldsymbol{H}, \boldsymbol{Z})$$

*Proof.* Consider $\mathcal{G} = (\boldsymbol{A}, \boldsymbol{X})$, where $\boldsymbol{X} \in \mathbb{R}^{N \times d}$ is derived from a latent graph $\mathcal{G}_{\mathcal{I}} = (\boldsymbol{A}, \boldsymbol{F})$ following a distribution $\mathcal{G} \sim P(\mathcal{G}_{\mathcal{I}})$, with $\boldsymbol{F} \in \mathbb{R}^{N \times d}$ representing the latent node semantics. Let $\hat{\mathcal{G}} = (\hat{\boldsymbol{X}}, \hat{\boldsymbol{A}})$ denote the randomly augmented version of $\mathcal{G}$, achieved by applying augmentations to both node features and graph structure. Additionally, consider an encoder $\mathcal{E}$ and a decoder $\mathcal{D}$ implemented as fully-connected layers, ensuring $l$-Lipschitz continuity with respect to the $l_2$-norm, and a non-parametric aggregator $\phi$. This yields $\hat{\boldsymbol{H}} = \mathcal{E}(\hat{\boldsymbol{X}})$ with $\hat{\boldsymbol{H}} \in \mathbb{R}^{N \times d'}$, and $\hat{\boldsymbol{Z}} = \phi(\hat{\boldsymbol{H}}, \hat{\boldsymbol{A}})$ with $\hat{\boldsymbol{Z}} \in \mathbb{R}^{N \times d'}$. Furthermore, $\boldsymbol{F}^* \in \mathbb{R}^{N \times d'}$ denotes the lossless compression of $\boldsymbol{F}$ that $\mathbb{E}[\boldsymbol{X}|\boldsymbol{A}, \boldsymbol{F}^*] = \boldsymbol{F}$. The Eq. 6 can be rewrited as:

$$\mathcal{E}^* = \arg\min_{\mathcal{E}} \mathbb{E}_{\hat{A},\hat{X}} \left[ \|\hat{H} - \hat{Z}\|^2 + \|\mathcal{D}(\hat{Z}) - X\|^2 | \hat{A}, \hat{X} \right] \tag{8}$$

$$= \arg\min_{\mathcal{E}} \mathbb{E}_{\hat{A},\hat{X}} \left[ \|(\hat{H} - F^*) - (\hat{Z} - F^*)\|^2 + \|\mathcal{D}(\hat{Z}) - X\|^2 | \hat{A}, \hat{X} \right] \tag{9}$$

$$= \arg\min_{\mathcal{E}} \mathbb{E}_{\hat{A},\hat{X}} \left[ \|\hat{H} - F^*\|^2 + \|\hat{Z} - F^*\|^2 + \|\mathcal{D}(\hat{Z}) - X\|^2 | \hat{A}, \hat{X} \right]$$
$$- 2\mathbb{E}_{\hat{A},\hat{X}} \left[ \langle \hat{H} - F^*, \hat{Z} - F^* \rangle | \hat{A}, \hat{X} \right] \tag{10}$$

$$= \arg\min_{\mathcal{E}} \mathbb{E}_{\hat{A},\hat{X}} \left[ \|\hat{H} - F^*\|^2 + \|\hat{Z} - F^*\|^2 + \|\mathcal{D}(\hat{Z}) - X\|^2 | \hat{A}, \hat{X} \right]$$
$$- 2\mathbb{E}_{\hat{A},\hat{X},F^*} \left[ \sum_i (\hat{H}_i - F_i^*)(\hat{Z}_i - F_i^*) | \hat{A}, \hat{X}, F^* \right] \tag{11}$$

$$= \arg\min_{\mathcal{E}} \mathbb{E}_{\hat{A},\hat{X}} \left[ \|\hat{H} - F^*\|^2 + \|\hat{Z} - F^*\|^2 + \|\mathcal{D}(\hat{Z}) - X\|^2 | \hat{A}, \hat{X} \right]$$
$$- 2\mathbb{E}_{\hat{A},\hat{X},F^*} \left[ \sum_i \mathrm{Cov}(\hat{H}_i - F_i^*, \hat{Z}_i - F_i^*) | \hat{A}, \hat{X}, F^* \right] \tag{12}$$

$$= \arg\min_{\mathcal{E}} \mathbb{E}_{\hat{A},\hat{X}} \left[ \|\hat{H} - F^*\|^2 + \|\hat{Z} - F^*\|^2 + \|\mathcal{D}(\hat{Z}) - X\|^2 | \hat{A}, \hat{X} \right]$$
$$- 2\mathbb{E}_{\hat{A},\hat{X},F^*} \left[ \sum_i \mathrm{Cov}(\hat{H}_i, \hat{Z}_i) | \hat{A}, \hat{X}, F^* \right]. \tag{13}$$

The terms $\|\hat{H} - F^*\|^2$ and $\|\hat{Z} - F^*\|^2$ represent the reconstruction errors of MLP representations $\hat{H}$ and GNN representations $\hat{Z}$ on the latent graph $\mathcal{G}_{\mathcal{I}}$. These errors ensure the invariance of these representations with respect to latent semantics. The term $\|\mathcal{D}(\hat{Z}) - X\|^2$ optimizes the distance between the reconstructed GNN representations and the node features. This optimization prevents representation shifts caused by augmentations. The final term, $-\sum_i \mathrm{Cov}(\hat{H}_i, \hat{Z}_i)$, quantifies the covariance between the representations encoded by MLPs and GNNs. Maximizing this covariance ensures that the encoder $\mathcal{E}$ effectively captures structural knowledge.

We will now interpret SSL-GM from the perspective of the information bottleneck. The fundamental idea behind the information bottleneck (Tishby et al., 2000; Tishby & Zaslavsky, 2015; Shwartz-Ziv & Tishby, 2017) is to compress the original information while retaining the latent information. In our case, the information to be compressed is the original graph $\mathcal{G}$, while we consider $\mathcal{G}_{\mathcal{I}}$ as the latent information. We regard the compressed information as the combination of MLP and GNN representations, denoted as $T = (H, Z)$. Thus, we define the information bottleneck as $T^* = \arg\min_T I(\mathcal{G}; T) - \beta I(T; \mathcal{G}_{\mathcal{I}})$. To present the information bottleneck in a more accessible manner, we perform the following transformation:

$$T^* = \arg\min_T I(\mathcal{G}; T) - \beta I(T; \mathcal{G}_{\mathcal{I}}) \tag{14}$$

$$= \arg\min_T (1 - \beta) H(T) + \beta H(T|\mathcal{G}_{\mathcal{I}}) - H(T|\mathcal{G}) \tag{15}$$

$$= \arg\min_T H(T) + \lambda H(T|\mathcal{G}_{\mathcal{I}}) \tag{16}$$

$$= \arg\min_{H,Z} \lambda H(H|\mathcal{G}_{\mathcal{I}}) + \lambda H(Z|H, \mathcal{G}_{\mathcal{I}}) + H(Z) + H(H|Z), \tag{17}$$

where $\lambda = \frac{\beta}{1-\beta} > 0$. We consider that the four terms in Eq. 17 correspond to the four terms in Eq. 13. Specifically, minimizing the conditional entropy $H(H|\mathcal{G}_{\mathcal{I}})$ and $H(Z|H, \mathcal{G}_{\mathcal{I}})$ in Eq. 17 can reduce the uncertainty of $H$ and $Z$ with respect to the latent graph $\mathcal{G}_{\mathcal{I}}$, thereby ensuring the invariance of $H$ and $Z$ concerning $\mathcal{G}_{\mathcal{I}}$. Consequently, these two terms correspond to the reconstruction terms $\|\hat{H} - F^*\|^2$ and $\|\hat{Z} - F^*\|^2$ in Eq. 13, respectively. Additionally, we posit that optimizing the third term $H(Z)$ in Eq. 17 reduces the uncertainty of $Z$. This aligns with the third term $\|\mathcal{D}(\hat{Z}) - X\|^2$

in Eq. 13, which constrains $Z$ to be invariant to $X$. Furthermore, concerning the final term in Eq. 17, which minimizes the conditional entropy $H(H|Z)$, it guarantees the consistency between MLP representations $H$ and GNN representations $Z$. This alignment can be attained by maximizing the covariance term $\sum_i \text{Cov}(\hat{H}_i, \hat{Z}_i)$, corresponding to the last term in Eq. 13. This analysis reveals that minimizing our objective is equivalent to optimizing the information bottleneck.

## B  EXPERIMENT SETUP DETAILS

In this section, we provide a comprehensive description of the experimental setup for both node classification and graph classification. The experiments are conducted on Nvidia A100 (80GB) for the Arxiv dataset, and Nvidia GeForce RTX 3090 (24GB) for the remaining datasets.

### B.1  DATASET STATISTICS

| Dataset | Task | # Graphs | # Nodes | # Edges | # Features | # Classes | Split |
|---|---|---|---|---|---|---|---|
| Cora | Node-level | 1 | 2,708 | 10,556 | 1,433 | 7 | 10%/10%/80% |
| Citeseer | Node-level | 1 | 3,327 | 9,104 | 3,703 | 6 | 10%/10%/80% |
| PubMed | Node-level | 1 | 19,717 | 88,648 | 500 | 3 | 10%/10%/80% |
| Amazon-CS | Node-level | 1 | 13,752 | 491,722 | 767 | 10 | 10%/10%/80% |
| Amazon-Photo | Node-level | 1 | 7,650 | 238,162 | 745 | 8 | 10%/10%/80% |
| Co-CS | Node-level | 1 | 18,333 | 163,788 | 6,805 | 15 | 10%/10%/80% |
| Co-Phys | Node-level | 1 | 34,493 | 495,924 | 8,415 | 5 | 10%/10%/80% |
| Wiki-CS | Node-level | 1 | 11,701 | 432,246 | 300 | 10 | 10%/10%/80% |
| Flickr | Node-level | 1 | 89,250 | 899,756 | 500 | 7 | 10%/10%/80% |
| Arxiv | Node-level | 1 | 169,343 | 1,166,243 | 128 | 40 | Public Split |

| Dataset | Task | # Graphs | # Nodes | # Edges | # Features | # Classes | Split |
|---|---|---|---|---|---|---|---|
| IMDB-B | Graph-level | 1,000 | ∼19.8 | ∼193.1 | - | 2 | 10-fold CV |
| IMDB-M | Graph-level | 1,500 | ∼13.0 | ∼65.9 | - | 3 | 10-fold CV |
| COLLAB | Graph-level | 5,000 | ∼74.5 | ∼4,914.4 | - | 3 | 10-fold CV |
| PTC-MR | Graph-level | 344 | ∼14.3 | ∼14.7 | 18 | 2 | 10-fold CV |
| MUTAG | Graph-level | 118 | ∼17.9 | ∼39.6 | 7 | 2 | 10-fold CV |
| DD | Graph-level | 1,178 | ∼284.3 | ∼715.6 | 89 | 2 | 10-fold CV |
| PROTEINS | Graph-level | 1,113 | ∼39.1 | ∼145.6 | 3 | 2 | 10-fold CV |

Table 6: The statistics of datasets for node-level and graph-level tasks.

We select 17 benchmark datasets, with 10 designated for node classification and 7 for graph classification, to evaluate the performance of SSL-GM and other approaches. These datasets are collected from diverse domains, encompassing citation networks, social networks, molecule networks, etc. We present the statistics of these datasets in Table 6.

**Node Classification**: Specifically, **Cora, Citeseer, PubMed** (Yang et al., 2016) are three citation networks, in which nodes denote papers and edges represent citations. The node features are represented as bag-of-words based on paper keywords. **Amazon-CS and Amazon-Photo** (Shchur et al., 2018) are two co-purchase networks that describe the frequent co-purchases of items (nodes). **Co-CS (Coauthor-CS) and Co-Phys (Coauthor-physics)** (Shchur et al., 2018) consist of nodes representing authors and edges indicating collaborations between authors. **Wiki-CS** (Mernyei & Cangea, 2020) is extracted from Wikipedia, comprising computer science articles (nodes) connected by hyperlinks (edges). **Flickr** (Zeng et al., 2020) consists online images, with the goal of categorizing images based on their descriptions and common properties. All these datasets are available through PyG (Pytorch Geometric), and we partition them randomly into training, validation, and testing sets with a split ratio of 10%/10%/80%. Additionally, we employ **Arxiv** dataset from OGB benchmarks (Hu et al., 2020) to evaluate model performance on large-scale datasets. We process the dataset in PyG using OGB public interfaces with standard public split setting.

**Graph Classification**: All graph classification datasets are sourced from TU datasets (Morris et al., 2020). We employ several datasets, including biochemical molecule datasets (**PTC-MR, MUTAG, DD, PROTEINS**) and social networks (**IMDB-B, IMDB-M, COLLAB**), to access whether SSL-GM can acquire generalizable and global information. In the case of the PTC-MR and DD datasets, we utilize the original node features, whereas for other datasets lacking rich node features, we generate one-hot features based on node degrees. These datasets are available in PyG library following a 10-fold cross validation data split.

## B.2 SUMMARY OF BASELINES

We compare SSL-GM against a range of baselines, encompassing supervised GNNs, self-supervised graph contrastive learning (GCL) methods, and MLP-based graph learning methods.

**Supervised GNNs.** Our primary node classification baselines include **GraphSAGE** (Hamilton et al., 2017) and **GAT** (Veličković et al., 2018), while for graph classification, we utilize **GIN** (Xu et al., 2019). Furthermore, we also incorporate **SGC** (Wu et al., 2019) and **APPNP** (Gasteiger et al., 2019) as additional node classification baselines.

**Self-supervised GNNs.** We compare SSL-GM to self-supervised graph learning methods. **DGI** (Veličković et al., 2019) and **MVGRL** (Hassani & Khasahmadi, 2020) conduct contrastive learning between graph patches and graph summaries to integrate knowledge into node representations. **GRACE** (Zhu et al., 2020) and subsequent **GCA** (Zhu et al., 2021) perform contrast between nodes in two corrupted views to acquire augmentation-invariant representations. **BGRL** (Thakoor et al., 2022) utilizes predictive objective for node-level contrastive learning to achieve efficient training. For graph-level tasks, we explore traditional graph kernels for classification, including **WL kernel** (Shervashidze et al., 2011) and **DGK** (Yanardag & Vishwanathan, 2015). Furthermore, we include contrastive learning approaches, such as **graph2vec** (Narayanan et al., 2017), **MVGRL** (Hassani & Khasahmadi, 2020), **InfoGraph** (Sun et al., 2020), **GraphCL** (You et al., 2020), and **JOAO** (You et al., 2021), which conduct contrastive learning between representations of two augmented graphs.

**MLPs on Graphs.** In node classification, we employ basic **MLP** that considers only node content as baseline. Furthermore, we incorporate **GraphMLP** (Hu et al., 2021) that trains an MLP by emphasizing consistency between target nodes and their direct neighborhoods. We exclude the following works (Dong et al., 2022; Liu et al., 2022) as baselines since they are high-order versions of GraphMLP. To achieve this, we slightly modify the original GraphMLP to enable the ability in learning high-order information, and search the number of layers within {1, 2, 3}. **GLNN** (Zhang et al., 2022) employs knowledge distillation to transfer knowledge from GNNs to MLPs, **GENN** leverages positional encoding to acquire structural knowledge, while **NOSMOG** (Tian et al., 2023) jointly integrates positional information and robust training strategies based on GLNN. Note that the public code of GENN is not available, thus we implement GENN based on the code of NOSMOG. In graph classification, we employ a pooling function to generate graph-level representations for training an MLP. For other baselines, they cannot be readily applied to graph-level tasks. Therefore, we implement an MLP with graph-level knowledge distillation as another baseline to access the role of knowledge distillation on graph classification.

## B.3 HYPER-PARAMETER SETTING

| Hyper-parameters | Node Classification | | | | | | | | | |
|---|---|---|---|---|---|---|---|---|---|---|
| | Cora | Citeseer | PubMed | Amazon-CS | Amazon-Photo | Co-CS | Co-Phys | Wiki-CS | Flickr | Arxiv |
| Epochs | 1000 | 1000 | 1000 | 1000 | 1000 | 2000 | 1000 | 2000 | 2000 | 5000 |
| Optimizer | | | | AdamW used for all datasets | | | | | | |
| Learning Rate | 1e-3 | 5e-4 | 5e-4 | 1e-3 | 1e-3 | 1e-4 | 1e-3 | 5e-4 | 1e-3 | 1e-3 |
| Weight Decay | 0 | 5e-5 | 1e-5 | - | 1e-4 | - | 1e-4 | 1e-5 | 5e-4 | - |
| Activation | | | | PReLU used for all datasets | | | | | | |
| Hidden Dimension | 512 | 512 | 512 | 512 | 512 | 512 | 512 | 512 | 1024 | 1024 |
| Normalization | | | | Batchnorm used for all datasets | | | | | | |
| # Encoder Layers | 2 | 2 | 2 | 3 | 2 | 2 | 2 | 2 | 2 | 8 |
| # Aggregator Layers | 2 | 3 | 3 | 2 | 1 | 1 | 1 | 2 | 3 | 3 |
| Feature Mask Ratio | 0.50 | 0.75 | 0.25 | 0.25 | 0.25 | 0.50 | 0.75 | 0.00 | 0.25 | 0.00 |
| Edge Mask Ratio | 0.25 | 0.50 | 0.25 | 0.25 | 0.50 | 0.75 | 0.50 | 0.25 | 0.50 | 0.25 |

Table 7: Hyper-parameters used for SSL-GM for node-level task.

We perform hyper-parameter tuning for each approach using a grid search strategy. Specifically, we set the number of epochs to 1,000, the hidden dimension to 512, and employ PReLU as the activation function. We explore various learning rates {5e-4, 1e-4, 5e-4, 1e-3, 5e-3, 1e-2}, weight decay values {5e-5, 1e-5, 5e-3, 1e-4, 0}, and the number of layers {1, 2, 3}. In self-supervised learning methods, we employ a 2-layer GCN (Kipf & Welling, 2017) as the encoder for node-level tasks. For graph-level tasks, we utilize a 5-layer GIN (Xu et al., 2019) model concatenated with a readout function, which is selected from {MEAN, SUM, MAX}. Subsequently, we assess the quality of the acquired representations by training a Logistic regression function on downstream

| Hyper-parameters | Graph Classification | | | | | | |
|---|---|---|---|---|---|---|---|
| | IMDB-B | IMDB-M | COLLAB | PTC-MR | MUTAG | DD | PROTEINS |
| Epochs | 200 | 100 | 30 | 100 | 100 | 100 | 500 |
| Optimizer | AdamW used for all datasets | | | | | | |
| Learning Rate | 1e-2 | 1e-2 | 5e-4 | 1e-2 | 1e-2 | 1e-3 | 1e-3 |
| Weight Decay | 0 used for all datasets | | | | | | |
| Activation | PReLU used for all datasets | | | | | | |
| Batch Size | 64 | 128 | 32 | 64 | 64 | 32 | 64 |
| Raw Feature | N | N | N | Y | N | Y | N |
| Deg4Feature | Y | Y | Y | N | Y | N | Y |
| Pooling | MEAN | MEAN | MEAN | SUM | SUM | MEAN | SUM |
| Hidden Dimension | 512 used for all datasets | | | | | | |
| Normalization | Batchnorm used for all datasets | | | | | | |
| # Encoder Layers | 2 used for all datasets | | | | | | |
| # Aggregator Layers | 2 | 2 | 2 | 2 | 1 | 2 | 1 |
| Feature Mask Ratio | 0.50 | 0.25 | 0.75 | 0.25 | 0.5 | 0.00 | 0.00 |
| Edge Mask Ratio | 0.75 | 0.50 | 0.75 | 0.00 | 0.25 | 0.00 | 0.50 |

Table 8: Hyper-parameters used for SSL-GM for graph-level task.

tasks (Zhu et al., 2020). For other settings, we follow the settings reported in the original papers. Regarding SSL-GM, we provide a comprehensive overview of the hyper-parameter settings for both node-level and graph-level tasks in Tables 7 and 8, respectively.

### B.4 DETAILED EVALUATION SETTING

**Transductive Setting.** We consider a graph $\mathcal{G} = (\mathcal{V}, E)$ in which all nodes are visible during the training stage. We partition the nodes into three non-overlapping sets: $\mathcal{V} = \mathcal{V}_{train} \sqcup \mathcal{V}_{val} \sqcup \mathcal{V}_{test}$. In the supervised setting, we train the encoder using $\mathcal{V}_{train}$ and evaluate its performance on $\mathcal{V}_{val}$ and $\mathcal{V}_{test}$. In the self-supervised setting, we train the encoder on $\mathcal{V}$ and utilize $\mathcal{V}_{trans}$ for training the downstream head, while using $\mathcal{V}_{val}$ and $\mathcal{V}_{test}$ for evaluation. We perform the split 10 times to evaluate the quality of the learned representations to alleviate the impact of randomness.

**Inductive (Production) Setting.** In the production setting, we partition a graph $\mathcal{G} = (\mathcal{V}, E)$ into transductive set $\mathcal{G}^T$ and inductive set $\mathcal{G}^I$, where $\mathcal{G} = \mathcal{G}^T \sqcup \mathcal{G}^I$ and $\emptyset = \mathcal{G}^T \sqcap \mathcal{G}^I$. $\mathcal{G}^T = (\mathcal{V}^T, E^T)$, containing 80% of the nodes, is used for training, while $\mathcal{G}^I = (\mathcal{V}^I, E^I)$, containing the remaining 20% of the nodes, remains unseen during training. We further partition the nodes in $\mathcal{G}^T$ into non-overlapping training, validation, and testing sets, denoted as $\mathcal{V}^T = \mathcal{V}^T_{train} \sqcup \mathcal{V}^T_{valid} \sqcup \mathcal{V}^T_{test}$. The model is trained on $\mathcal{G}^T$, and we report the transductive results on $\mathcal{V}^T_{test}$ and inductive results on $\mathcal{V}^I$. We interpolate the results of these two settings to represent production results. Our approach differs from that of Zhang et al. (2022) and Tian et al. (2023). We treat nodes in the inductive set as disconnected from nodes in the transductive set, even during inference, creating a more challenging out-of-distribution setting.

**Cold-start Setting.** The cold-start setting is closely similar to inductive (production) setting, but we assume the nodes in the inductive set are isolated, $\mathcal{G}^C = (\mathcal{V}^I, \emptyset)$. This is a more challenging yet practical setting, as new agents often emerge independently in real-world systems. We follow the production setting to train the model on transductive set $\mathcal{G}^T$ but only report the model performance on cold-start set $\mathcal{G}^C$ to assess cold-start performance. For models that rely on positional encoding, such as GENN and NOSMOG, we set the positional embeddings as zero vectors.

## C EMPIRICALLY RUNNING TIME AND MEMORY USAGE

Table 9 presents a comparison of the running time and memory usage between our SSL-GM and other baseline methods, namely GAT (Veličković et al., 2018), GRACE (Zhu et al., 2020), and BGRL (Thakoor et al., 2022). Despite our primary focus on accelerating inference speed, we have observed that SSL-GM outperforms these approaches in both training time and memory utilization. In particular, GAT, which employs 4 attention heads, imposes a substantial computational burden during training due to attention score computation, resulting in significant memory consumption. Considering self-supervised methods, GRACE utilizes the InfoNCE loss for model training, involving computation of node pair similarities. This operation results in considerable time and memory overhead. In comparison to GRACE, our SSL-GM demonstrates improvements in terms of memory

usage ($3.8 \sim 6.8\times$) and training time efficiency ($4.8 \sim 8.3\times$). BGRL employs bootstrap loss (Grill et al., 2020) to predict node representations from different views, remarkably enhancing training and memory efficiency. However, our SSL-GM remains more efficient than BGRL.

Additionally, we conducted comparison between SSL-GM and other acceleration techniques on Flickr and Arxiv datasets, as summarized in Table 10 and 11. Note that the inference time for self-supervised approaches in this paper aligns with that of SAGE (Hamilton et al., 2017). Therefore, we establish SAGE as our baseline and apply acceleration techniques based on that, including quantization (QSAGE), pruning (PSAGE), and neighbor sampling (Neighbor Sample), to facilitate the inference. Furthermore, we include an evaluation of the inference time of SGC (Wu et al., 2019), which comprises an MLP and a one-layer message passing. It is evident that even the most efficient methods achieve only modest acceleration ($3.2 \sim 4.0\times$), and SGC, which employs only a single-layer aggregation, achieves a mere acceleration ($1.1 \sim 1.2\times$). This observation demonstrates that the aggregation process leads to significant time consumption during inference. In contrast, our SSL-GM achieves remarkable inference acceleration by disregarding neighborhood dependency, which is faster than SAGE ($89.7 \sim 125.9\times$). Compared to other methods employing MLPs as encoders like GLNN and NOSMOG, we do not observe significant distinctions in terms of inference speed. However, methods like NOSMOG (Tian et al., 2023) and GENN (Wang et al., 2023) utilize additional positional embeddings, introducing significant time consumption in learning positional embedding. For example, NOSMOG remains fast inference under 5 milliseconds, while the positional encoding takes more than 5 seconds on the Arxiv dataset, even with a minimal epoch setting of 1. Considering the time consumption on encoding positional embeddings, the overall inference consumption of GENN and NOSMOG far surpasses that of SAGE.

| Dataset | Amazon-CS | | Amazon-Photo | | Coauthor-CS | | Coauthor-Phys | | Wiki-CS | |
|---|---|---|---|---|---|---|---|---|---|---|
| | Memory | Training Time | Memory | Training Time | Memory | Training Time | Memory | Training Time | Memory | Training Time |
| GAT | 5239 MB | 73.8 (s) | 2571 MB | 41.9 (s) | 2539 MB | 60.4 (s) | 13199 MB | 265.2 (s) | 4568 MB | 74.4 (s) |
| GRACE | 8142 MB | 349.5 (s) | 2755 MB | 138.4 (s) | 11643 MB | 261.4 (s) | 16294 MB | 573.2 (s) | 5966 MB | 290.9 (s) |
| BGRL | 2196 MB | 96.8 (s) | 1088 MB | 64.1 (s) | 2513 MB | 129.9 (s) | 5556 MB | 273.8 (s) | 1899 MB | 108.8 (s) |
| **SSL-GM** | **1969 MB** | **53.4 (s)** | **694 MB** | **27.0 (s)** | **1716 MB** | **54.8 (s)** | **3920 MB** | **110.7 (s)** | **1590 MB** | **35.5 (s)** |

Table 9: Computational requirements of different baseline methods on a set of standard benchmark graphs. The experiments are performed on a 24GB Nvidia GeForce RTX 3090.

| Datasets | | SAGE | BGRL | SGC | APPNP | QSAGE | PSAGE | Neighbor Sample | **SSL-GM** |
|---|---|---|---|---|---|---|---|---|---|
| Flickr | Time (ms) | 80.7 | 80.7 (1.00×) | 76.9 (1.05×) | 78.1 (1.03×) | 70.6 (1.14×) | 67.4 (1.20×) | 25.5 (3.16×) | **0.9 (89.67×)** |
| | Acc (%) | 47.17 | 49.12 | 47.35 | 47.53 | 47.22 | 47.25 | 47.01 | **49.27** |
| Arxiv | Time (ms) | 314.7 | 314.7 (1.00×) | 265.9 (1.18×) | 284.1 (1.11×) | 289.5 (1.09×) | 297.5 (1.06×) | 78.3 (4.02×) | **2.5 (125.88×)** |
| | Acc (%) | 68.52 | 69.29 | 68.93 | 69.10 | 68.48 | 68.55 | 68.35 | **70.23** |

Table 10: The inference time and accuracy of different acceleration methods.

| Dataset | Models | Trans | | Ind | | Cold-start | |
|---|---|---|---|---|---|---|---|
| | | Time (ms) | Acc (%) | Time (ms) | Acc (%) | Time (ms) | Acc (%) |
| Pubmed | SAGE | 73 | 85.94 | 15 | 85.04 | 15 | 77.98 |
| | SGC | 64 | 85.28 | 14 | 85.22 | 13 | 76.10 |
| | NOSMOG | 5 | 86.18 | 3 | 83.84 | 3 | 81.48 |
| | SSL-GM | **3** | **86.99** | **3** | **86.47** | **3** | **86.44** |
| Amazon-CS | SAGE | 103 | 88.88 | 31 | 87.24 | 25 | 61.01 |
| | SGC | 89 | **89.31** | 26 | 87.12 | 24 | 63.08 |
| | NOSMOG | 5 | 87.64 | 4 | 86.61 | 4 | 81.95 |
| | SSL-GM | **4** | 88.46 | **3** | **87.65** | **3** | **87.58** |
| Arxiv | SAGE | 485 | **72.05** | 315 | 68.52 | 305 | 43.47 |
| | SGC | 410 | 69.95 | 266 | 68.93 | 250 | 42.08 |
| | NOSMOG | 6 | 70.84 | 4 | 69.10 | 4 | 61.64 |
| | SSL-GM | **4** | 71.12 | **3** | **70.23** | **3** | **66.13** |

Table 11: Inference acceleration across different settings.

| | | IMDB-B | IMDB-M | COLLAB | PTC-MR | MUTAG | DD | PROTEINS |
|---|---|---|---|---|---|---|---|---|
| Supervised | GIN | 75.10±5.10 | 52.30±2.80 | 80.20±1.90 | 64.60±1.70 | 89.40±5.60 | 74.88±3.12 | 76.20±2.80 |
| Graph Kernel | WL | 72.30±3.44 | 46.95±0.46 | - | 57.97±0.49 | 80.72±3.00 | - | 72.92±0.56 |
| | DGK | 66.96±0.56 | 44.55±0.52 | - | 60.08±2.55 | 87.44±2.72 | - | 73.30±0.82 |
| GCL | graph2vec | 71.10±0.54 | 50.44±0.87 | - | 60.17±6.86 | 83.15±9.25 | - | 73.30±2.05 |
| | MVGRL | 71.84±0.78 | 50.84±0.92 | 73.10±0.56 | - | **89.24±1.31** | 75.20±0.55 | 74.02±0.32 |
| | InfoGraph | 73.03±0.87 | 49.69±0.53 | 70.65±1.13 | **61.65±1.43** | 89.01±1.13 | 72.85±1.78 | 74.44±0.31 |
| | GraphCL | 71.14±0.44 | 48.58±0.67 | 71.36±1.15 | - | 86.80±1.34 | **78.62±0.40** | 74.39±0.45 |
| | JOAO | 70.21±3.08 | 49.20±0.77 | 69.50±0.36 | - | 87.35±1.02 | - | 74.55±0.41 |
| MLP | MLP* | 49.50±1.66 | 33.11±1.59 | 51.90±0.95 | 54.39±1.41 | 67.22±0.99 | 58.56±1.40 | 59.20±1.00 |
| | MLP + KD* | 72.85±1.04 | 48.14±0.52 | 75.38±1.53 | 59.38±1.38 | 87.44±0.67 | 73.59±1.69 | 73.54±1.78 |
| **SSL-GM** | | **74.06±0.22** | **51.41±0.52** | **81.04±0.11** | 60.28±1.07 | 87.67±0.24 | 78.44±0.47 | **75.31±0.13** |
| $\Delta_{GraphCL}$ | | ↑ 4.10% | ↑ 5.83% | ↑ 13.57% | - | ↑ 1.00% | ↓ 0.23% | ↑ 1.24% |
| $\Delta_{MLP}$ | | ↑ 49.62% | ↑ 55.27% | ↑ 56.15% | ↑ 10.83% | ↑ 30.42% | ↑ 33.95% | ↑ 27.21% |
| $\Delta_{MLP+KD}$ | | ↑ 1.66% | ↑ 6.79% | ↑ 7.51% | ↑ 1.52% | ↑ 0.26% | ↑ 6.59% | ↑ 2.41% |

The reported results of baselines are from previous papers if available (You et al., 2020; 2021; Hou et al., 2022). * indicates the results are from our implementation.

Table 12: Graph classification accuracy (%). $\Delta_{MLP}, \Delta_{GraphCL}, \Delta_{MLP+KD}$ represents the performance gap (%) between our methods and GraphCL, MLP, and knowledge distillation-enhanced MLP, where green indicates the improvement over 4% and red indicates the degradation.

# D ADDITIONAL EXPERIMENTAL RESULTS

## D.1 EFFECTIVENESS OF SSL-GM ON GRAPH CLASSIFICATION

In this section, we present the experimental results of SSL-GM alongside state-of-the-art baselines for graph classification. The datasets encompass a variety of graph types, including biomolecular graphs and social networks (as described in Appendix B.1), and the baselines comprise supervised GNNs, traditional graph kernels, and self-supervised graph learning approaches (as detailed in Appendix B.2). Note that existing MLP-based methods, such as GLNN and NOSMOG, are specifically designed for node-level tasks, which hinder their direct applicability to graph classification tasks. To assess the impact of knowledge distillation on graph-level tasks, we also implement a combination of MLP and KD, which is similar to GLNN but conducts knowledge distillation at the graph level. Due to the generalizability inherent in self-supervised learning (Sun et al., 2023), SSL-GM can be readily extended to various downstream tasks. Our results demonstrate that SSL-GM outperforms other MLP-based methods, achieving the best or sub-best performance on 6 out of 7 baselines. In contrast, MLP-based methods generally underperform when compared to self-supervised GCL methods and even traditional graph kernel methods. Notably, SSL-GM surpasses other baselines, even on the large-scale dataset COLLAB, highlighting the potential of MLP-based methods in graph-level tasks. Our findings expand the usability of MLPs from node-level tasks to graph-level tasks.

## D.2 COMPREHENSIVE PREDICTION RESULTS UNDER INDUCTIVE (PRODUCTION) SETTING

Table 13 presents the comprehensive experimental results of our inductive (production) setting. We report the performance on both transductive and inductive sets, along with the interpolated production results. Note that the results of GLNN (Zhang et al., 2022) and NOSMOG (Tian et al., 2023) reported in this paper differ from their respective original papers. This discrepancy arises because our experimental setting presents a more challenging task where the inductive set is disconnected from the transductive set in inference. In this table, we present the performance of six baseline methods, which encompass supervised GNNs, self-supervised approaches, and MLP-based techniques. We can observe that our SSL-GM attains state-of-the-art performance in the majority of settings. Among GNN methods, we note that SAGE generally outperforms BGRL in transductive settings, but underperforms it in inductive settings. The robustness of BGRL stems from the augmentation, which aids in learning augmentation-invariant representations, enabling BGRL to work on inductive sets, even though the distributions of these two sets are potentially different. Regarding MLP-based methods, we observe that NOSMOG outperforms GLNN in transductive settings, particularly on large-scale graphs, while GLNN significantly outperforms NOSMOG in inductive settings. We assume that the learned positional embeddings of NOSMOG on the inductive set differ from those on the transductive set, rendering them untrustworthy and less meaningful. For our SSL-GM, the

model maximizes consistency of MLPs and GNNs in the representation space, thereby preserving more fine-grained structural knowledge, which ensures the model generalization.

| | Setting | Cora | Citeseer | PubMed | Amazon-CS | Amazon-Photo | Co-CS | Co-Phys | Wiki-CS | Flickr | Arxiv |
|---|---|---|---|---|---|---|---|---|---|---|---|
| SAGE | *prod* | 77.51±1.77 | 68.40±1.61 | 85.04±0.44 | 87.24±0.43 | 93.20±0.45 | 92.88±0.40 | 95.74±0.12 | **79.26±0.65** | 47.17±0.73 | 68.52±0.56 |
| | *trans* | 79.46±1.49 | 68.73±1.37 | 85.57±0.30 | 87.98±0.30 | 93.70±0.42 | 93.13±0.33 | 95.77±0.04 | 80.01±0.41 | 48.15±0.63 | **71.79±0.50** |
| | *ind* | 69.70±2.89 | 67.11±2.57 | 82.90±0.98 | 84.45±0.94 | 91.18±0.56 | 91.87±0.68 | 95.63±0.05 | 76.27±1.63 | 43.25±1.14 | 55.45±0.78 |
| BGRL | *prod* | 77.73±1.07 | 64.33±1.56 | 83.97±0.48 | 87.33±0.48 | 91.47±0.62 | 91.26±0.35 | 94.38±0.29 | 76.25±1.09 | 49.12±0.31 | 69.29±0.38 |
| | *trans* | 77.32±0.90 | 64.15±1.40 | 83.97±0.34 | 87.27±0.42 | 91.47±0.51 | 91.31±0.33 | 94.40±0.25 | 76.32±0.97 | 49.09±0.24 | 70.36±0.35 |
| | *ind* | 79.38±1.74 | 65.03±2.19 | 83.98±1.02 | **87.59±0.75** | 91.46±1.05 | 91.09±0.45 | 94.33±0.46 | 75.96±1.57 | 49.26±0.60 | 65.03±0.50 |
| MLP | *prod* | 63.76±1.65 | 63.98±1.22 | 80.91±0.45 | 81.00±0.54 | 87.73±0.88 | 91.68±0.59 | 95.18±0.13 | 75.08±0.71 | 46.14±0.22 | 55.89±0.51 |
| | *trans* | 63.66±1.53 | 63.86±1.09 | 80.92±0.38 | 81.05±0.45 | 87.69±0.86 | 91.66±0.54 | 95.18±0.12 | 75.12±0.43 | 46.16±0.15 | 55.89±0.46 |
| | *ind* | 64.15±2.11 | 64.43±1.76 | 80.90±0.72 | 80.80±0.91 | 87.88±0.96 | 91.78±0.81 | 95.16±0.18 | 74.94±1.81 | 46.09±0.50 | 55.91±0.69 |
| GLNN | *prod* | 78.34±1.04 | 69.61±1.13 | 85.44±0.48 | 87.04±0.50 | 93.28±0.43 | 93.72±0.35 | 95.76±0.09 | 78.39±0.54 | 46.11±0.27 | 63.53±0.48 |
| | *trans* | 79.93±0.87 | 69.73±0.77 | 85.70±0.38 | 87.80±0.45 | 93.84±0.42 | 93.82±0.32 | 95.78±0.04 | 78.58±0.32 | 46.13±0.22 | 64.27±0.46 |
| | *ind* | 71.96±1.68 | 69.14±2.58 | 84.42±0.87 | 83.98±0.70 | 91.05±0.49 | 93.34±0.47 | 95.70±0.09 | 77.64±1.42 | 46.05±0.43 | 60.55±0.55 |
| GENN | *prod* | 77.83±1.57 | 67.30±1.48 | 84.34±0.47 | 85.75±1.20 | 92.09±0.96 | 93.57±0.37 | 95.67±0.06 | 78.27±1.01 | 45.56±0.51 | 68.52±0.54 |
| | *trans* | 80.27±1.41 | 67.86±1.16 | 85.81±0.38 | 87.42±1.04 | 93.35±0.60 | 93.79±0.35 | 95.78±0.05 | 80.31±0.85 | 45.68±0.45 | 70.01±0.50 |
| | *ind* | 68.09±2.21 | 65.07±2.78 | 78.44±0.84 | 79.09±1.84 | 87.05±2.42 | 92.68±0.45 | 95.23±0.08 | 70.13±1.65 | 45.08±0.74 | 62.58±0.69 |
| NOSMOG | *prod* | 77.83±1.94 | 68.58±1.41 | 83.84±0.45 | 86.61±1.22 | 92.52±0.68 | 93.45±0.44 | 95.78±0.10 | 78.35±0.70 | 46.05±0.55 | 69.10±0.80 |
| | *trans* | 80.27±1.69 | 68.95±1.24 | 85.43±0.37 | **88.30±1.14** | **93.88±0.47** | 93.68±0.38 | 95.85±0.10 | **80.35±0.58** | 46.24±0.51 | 70.50±0.79 |
| | *ind* | 68.11±2.95 | 67.07±2.10 | 77.44±0.80 | 79.83±1.52 | 87.08±1.52 | 92.55±0.69 | 95.52±0.10 | 70.36±1.18 | 45.27±0.72 | 63.49±0.83 |
| **SSL-GM** | *prod* | **81.37±1.20** | **72.33±0.90** | **86.47±0.28** | **87.65±0.40** | **93.87±0.32** | **94.63±0.16** | **96.04±0.12** | 79.26±0.83 | **49.27±0.18** | **70.23±0.47** |
| | *trans* | **81.60±0.96** | **72.21±0.73** | **86.48±0.23** | 87.67±0.25 | 93.86±0.26 | **94.66±0.16** | **96.06±0.09** | 79.46±0.66 | **49.23±0.11** | 71.25±0.33 |
| | *ind* | **80.48±2.15** | **72.81±1.61** | **86.44±0.51** | 87.58±0.99 | **93.91±0.58** | **94.51±0.15** | **95.97±0.24** | **78.46±1.48** | **49.41±0.46** | **66.13±1.05** |

Table 13: Node classification accuracy (%) under inductive (production) scenario for both transductive and inductive settings. *ind* represents the accuracy on $\mathcal{V}^I$, *trans* represents the accuracy on $\mathcal{V}^T_{test}$, and *prod* is the interpolated accuracy of both *ind* and *trans*.

### D.3 EMPLOYING STANDARD GNNS WILL LEAD TO MODEL COLLAPSE

In this section, we attempt to answer why we approximate the representations of GNNs by a non-parametric aggregator instead of standard GNNs. We empirically demonstrate that the use of GNNs will lead to model collapse, which is a common issue in contrastive learning. To solve the issue, existing works in computer vision employ moving average network (Grill et al., 2020) to preserve the consistency between representations learned from two networks. Inspired by this, we propose a non-parametric aggregator to approximate the GNN representations based on MLP representations, thereby preserving the inherenet consistency between these two representations. In Figure 5, we compare the accuracy and loss curves between two versions of SSL-GM: one utilizes the non-parametric aggregator (MLP + PROP.), and the other directly employs GNNs. The experiments are conducted on five benchmark datasets in a transductive setting. We observe that MLP + PROP. achieves superior accuracy and higher losses than the GNN version. Specifically, we suppose the model collapse stems from the lack of consistency between the outputs of GNNs and MLPs in the representation space, which is a significant problem in contrastive learning (He et al., 2020). In our SSL-GM, we utilize the non-parametric aggregator to approximate GNN representations based on MLP representations, naturally preserving the consistency between MLPs and GNNs in the representation space. Our observation aligns with the findings in (He et al., 2020; Thakoor et al., 2022).

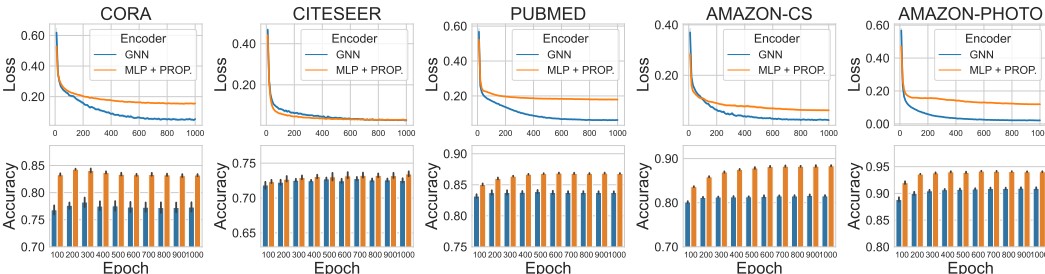

Figure 5: The training curves and model accuracies on five benchmark datasets. We compare the method applied in our SSL-GM (MLP + PROP.) with standard GNNs in learning node representations. We observe that employing GNNs will lead to model collapse.

## D.4 HOW AUGMENTATIONS IMPROVE THE MODEL CAPABILITY?

In this section, we evaluate the influence of augmentations on model performance. We conduct an ablation study to assess the model performance without specific augmentations, demonstrating both semantic and structural augmentations enhance model performance. Table 14 presents a comprehensive ablation study of edge masking and node feature masking for node classification on ten benchmark datasets within a transductive setting. We observe that these two types of augmentations significantly enhance model performance by improving different aspects of the datasets. Furthermore, the combination of these two techniques further enhance the performance of SSL-GM, indicating that our model can benefit from both augmentations simultaneously.

Additionally, we conduct a detailed analysis of how augmentations impact model performance. Figure 6 illustrates the model performance at different augmentation probabilities on Cora, Citeseer, PubMed, Amazon-CS, and Amazon-Photo datasets within a transductive setting. The augmentation ratio is searched among $\{0.0, 0.25, 0.5, 0.75\}$. These figures enable us to gain insight into the specific effects of augmentations on model performance.

| Feature Masking | Edge Masking | Cora | Citeseer | PubMed | Amazon-CS | Amazon-Photo | Co-CS | Co-Phys | Wiki-CS | Flickr | Arxiv |
|---|---|---|---|---|---|---|---|---|---|---|---|
| - | - | $82.10_{\pm0.45}$ | $71.83_{\pm0.43}$ | $86.89_{\pm0.13}$ | $87.12_{\pm0.15}$ | $93.52_{\pm0.20}$ | $93.10_{\pm0.05}$ | $94.56_{\pm0.06}$ | $80.98_{\pm0.13}$ | $48.21_{\pm0.10}$ | $70.58_{\pm0.20}$ |
| √ | - | $84.78_{\pm0.25}$ | $73.00_{\pm0.63}$ | $86.98_{\pm0.09}$ | $88.27_{\pm0.18}$ | $94.19_{\pm0.14}$ | $94.50_{\pm0.10}$ | $96.12_{\pm0.06}$ | $81.03_{\pm0.11}$ | $49.55_{\pm0.11}$ | $70.03_{\pm0.23}$ |
| - | √ | $82.33_{\pm0.61}$ | $71.78_{\pm0.77}$ | $86.98_{\pm0.13}$ | $87.35_{\pm0.29}$ | $93.69_{\pm0.07}$ | $94.35_{\pm0.08}$ | $95.88_{\pm0.06}$ | $81.04_{\pm0.22}$ | $49.33_{\pm0.07}$ | $\mathbf{71.12_{\pm0.10}}$ |
| √ | √ | $\mathbf{84.60_{\pm0.24}}$ | $\mathbf{73.52_{\pm0.53}}$ | $\mathbf{86.99_{\pm0.09}}$ | $\mathbf{88.46_{\pm0.16}}$ | $\mathbf{94.28_{\pm0.08}}$ | $\mathbf{94.87_{\pm0.07}}$ | $\mathbf{96.17_{\pm0.03}}$ | $\mathbf{81.21_{\pm0.13}}$ | $\mathbf{49.85_{\pm0.09}}$ | $\mathbf{71.12_{\pm0.10}}$ |

Table 14: Ablation study of augmentation methods used in SSL-GM.

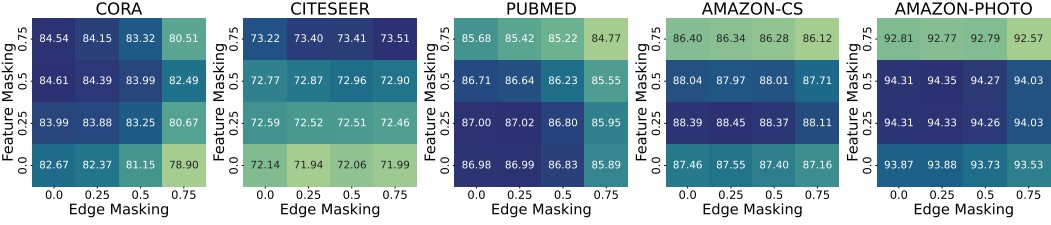

Figure 6: Node classification accuracy on transductive setting with different augmentation ratios.

## D.5 DOES THE RECONSTRUCTION TERM PREVENT REPRESENTATION SHIFTS?

In this section, we evaluate the role of the reconstruction term in SSL-GM. It is important to note that the reconstruction term serves to mitigate representation shifts caused by augmentation. The reconstruction term enables the outputs of GNNs to remain invariant to node features, thereby preserving more localized knowledge. Intuitively, if we do not apply augmentations, the incorporation of the reconstruction term will hinder the process of contrastive learning. To evaluate this, we assess the impact of the reconstruction term with and without applying augmentations, shown in Table 15. Intriguingly, the model performance demonstrates that the reconstruction term is effective only when augmentation is utilized. Without augmentation, the model performance is adversely affected. When both augmentation and reconstruction are applied, our SSL-GM achieves the highest performance, demonstrating the importance of employing reconstruction to prevent representation shifts brought by augmentation.

| Aug. | Rec. | Cora | Citeseer | PubMed | Amazon-CS | Amazon-Photo | Co-CS | Co-Phys | Wiki-CS | Flickr | Arxiv |
|---|---|---|---|---|---|---|---|---|---|---|---|
| - | - | $\mathbf{82.80_{\pm0.65}}$ | $\mathbf{72.03_{\pm0.83}}$ | $\mathbf{86.89_{\pm0.07}}$ | $\mathbf{87.42_{\pm0.17}}$ | $\mathbf{93.87_{\pm0.04}}$ | $\mathbf{93.12_{\pm0.03}}$ | $94.55_{\pm0.04}$ | $\mathbf{81.06_{\pm0.13}}$ | $\mathbf{48.61_{\pm0.10}}$ | $70.38_{\pm0.32}$ |
| - | √ | $82.10_{\pm0.45}$ | $71.83_{\pm0.43}$ | $\mathbf{86.89_{\pm0.13}}$ | $87.12_{\pm0.15}$ | $93.52_{\pm0.20}$ | $93.10_{\pm0.05}$ | $\mathbf{94.56_{\pm0.06}}$ | $80.98_{\pm0.13}$ | $48.21_{\pm0.10}$ | $\mathbf{70.58_{\pm0.20}}$ |
| √ | √ | $84.60_{\pm0.24}$ | $73.52_{\pm0.53}$ | $86.99_{\pm0.09}$ | $88.46_{\pm0.16}$ | $94.28_{\pm0.08}$ | $94.87_{\pm0.07}$ | $96.17_{\pm0.03}$ | $81.21_{\pm0.13}$ | $49.85_{\pm0.09}$ | $71.12_{\pm0.10}$ |

Table 15: Reconstruction term leads to performance drop without augmentation.

D.6 INFLUENCE OF AGGREGATOR TYPES AND AGGREGATION LAYERS

In this section, we examine the role of the aggregator of SSL-GM. Unlike SGC or APPNP, which employ high-order adjacent matrices to guide the message passing, we utilize a GNN-like architecture to guide aggregation, preserving non-linearity between layers. While various aggregation methods can be employed, we adopt the normalized Laplacian matrix, which is similar to GCN (Kipf & Welling, 2017), to aggregate high-order representations, as formulated as:

$$\boldsymbol{H}^{(l)} = \phi^{(l)}(\boldsymbol{A}, \boldsymbol{H}^{(l-1)}) = \sigma(\hat{\boldsymbol{D}}^{-\frac{1}{2}}\hat{\boldsymbol{A}}\hat{\boldsymbol{D}}^{-\frac{1}{2}}\boldsymbol{H}^{(l-1)}) \tag{18}$$

where $\boldsymbol{H}^{(l)}$ represents the high-order representation at $l$-th aggregator layer, $\phi^{(l)}(\cdot)$ denotes the $l$-th aggregator, $\hat{\boldsymbol{A}} = \boldsymbol{A} + \boldsymbol{I}$ is adjacent matrix with self-loop, $\hat{\boldsymbol{D}}$ denotes the degree matrix of $\hat{\boldsymbol{A}}$, and $\sigma$ is the activation function. The aggregation process is iterated $L$ times to yield the final high-order representations $\boldsymbol{Z} = \phi^{(L)}(\boldsymbol{A}, \boldsymbol{H}^{(L-1)})$. Note that the aggregation is non-parametric.

Additionally, we explore two other aggregation types, the row-normalized Laplacian matrix $\tilde{\boldsymbol{A}}\tilde{\boldsymbol{D}}^{-1}$ and column-normalized Laplacian matrix $\tilde{\boldsymbol{D}}^{-1}\tilde{\boldsymbol{A}}$. Table 16 presents the results of these three aggregation types. In this table, we observe that there is no significant difference in performance among the various aggregation methods. All of these methods can achieve desirable performance. Nevertheless, the GCN-like aggregation method consistently outperforms the others. We consider that if the encoder is non-parametric, then different aggregation methods will not bring significant differences in inductive bias, aligning with the findings in Yang et al. (2023a).

| | Cora | Citeseer | PubMed | Amazon-CS | Amazon-Photo | Co-CS | Co-Phys | Wiki-CS | Flickr | Arxiv |
|---|---|---|---|---|---|---|---|---|---|---|
| GCN (Ours) | **84.60**±0.24 | **73.52**±0.53 | **86.99**±0.09 | **88.46**±0.16 | **94.28**±0.08 | **94.87**±0.07 | **96.17**±0.03 | **81.21**±0.13 | **49.85**±0.09 | **71.12**±0.10 |
| Col | 84.14±0.34 | 73.48±0.53 | 86.92±0.08 | 87.93±0.27 | 93.11±0.15 | 94.81±0.06 | 96.09±0.03 | 80.62±0.30 | 49.15±0.16 | 71.03±0.09 |
| Row | 84.09±0.32 | 73.49±0.54 | 86.92±0.08 | 87.96±0.27 | 93.07±0.15 | 94.82±0.06 | 96.07±0.04 | 80.63±0.25 | 49.18±0.10 | 71.04±0.09 |

Table 16: Ablation study on node aggregation type. *GCN* indicates bi-normalized Laplacian aggregation matrix $\tilde{\boldsymbol{D}}^{-1/2}\tilde{\boldsymbol{A}}\tilde{\boldsymbol{D}}^{-1/2}$, *Col* indicates column-normalized Laplacian aggregation matrix $\tilde{\boldsymbol{D}}^{-1}\tilde{\boldsymbol{A}}$, and *Row* indicates row-normalized Laplacian aggregation matrix $\tilde{\boldsymbol{A}}\tilde{\boldsymbol{D}}^{-1}$.

In addition to analyzing the aggregation type, we also assess the performance of SSL-GM with varying numbers of aggregation layers, as depicted in Figure 7. In this figure, we illustrate the performance of SSL-GM on five benchmark datasets for both validation and testing sets under the transductive setting. We can see that the model attains the optimal performance with 2 or 3 layers, consistent with prior research on GNNs (Kipf & Welling, 2017; Hamilton et al., 2017). We consider that our model might encounter over-smoothing issues with a high number of layers (Li et al., 2019).

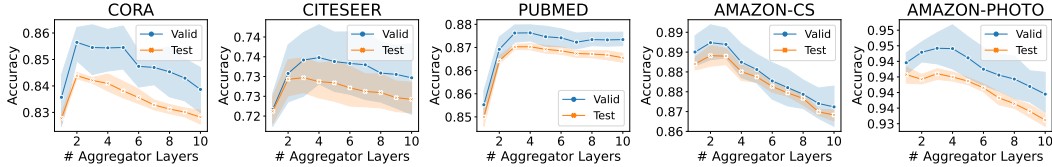

Figure 7: Node classification accuracy on transductive setting with different aggregation layers.

E EXTENSION TO SUPERVISED SSL-GM

Our SSL-GM is a self-supervised method that follows the training then evaluation paradigm. In particular, the model initially generates representations for each node and subsequently trains classification heads for downstream tasks. Despite the generalization capability of self-supervised learning techniques, their performance typically lags behind that of supervised methods (Chen et al., 2020b). To bridge the gap, we implement a supervised version of SSL-GM by jointly optimizing the objective function and downstream task, which can be formulated as

$$\mathcal{L} = \mathcal{L}_{cont} + \lambda \cdot \mathcal{L}_{rec} + \beta \cdot \mathcal{L}_{sup}. \tag{19}$$

We evaluate the performance of supervised SSL-GM on transductive node classification and search the value of $\beta$ within the range $\{0.1, 1, 10\}$. Table 17 reports the experimental results. It is counter-intuitive that our self-supervised version outperforms the supervised version on 7 out of 10 datasets. We suppose that the integration of cross-entropy loss may impact the learning process of contrastive learning by introducing additional gradients on model parameters. This could lead the model to acquire pseudo-knowledge unrelated to our prediction target. In other words, the model parameters might be optimized by the annotations rather than the outputs of GNNs.

|  | Cora | Citeseer | PubMed | Amazon-CS | Amazon-Photo | Co-CS | Co-Phys | Wiki-CS | Flickr | Arxiv |
|---|---|---|---|---|---|---|---|---|---|---|
| w/o sup. | **84.60**$_{\pm0.24}$ | 73.52$_{\pm0.53}$ | 86.99$_{\pm0.09}$ | 88.46$_{\pm0.16}$ | **94.28**$_{\pm0.08}$ | **94.87**$_{\pm0.07}$ | **96.17**$_{\pm0.03}$ | **81.21**$_{\pm0.13}$ | **49.85**$_{\pm0.09}$ | **71.12**$_{\pm0.10}$ |
| w.sup. | 77.23$_{\pm1.41}$ | **75.20**$_{\pm0.55}$ | **88.64**$_{\pm0.27}$ | **88.68**$_{\pm0.25}$ | 94.17$_{\pm0.18}$ | 94.60$_{\pm0.19}$ | 95.89$_{\pm0.10}$ | 80.64$_{\pm0.34}$ | 46.39$_{\pm1.73}$ | 70.60$_{\pm0.43}$ |

Table 17: Ablation study on the model with or without supervisions.

