# OpenReview forum: "Graph Inference Acceleration by Bridging GNNs and MLPs with Self-Supervised Learning"
_ICLR.cc/2024/Conference — Submitted to ICLR 2024_

### Official Review · Reviewer_rTyJ · 2023-10-19

**Soundness:** 2 fair
**Presentation:** 3 good
**Contribution:** 2 fair
**Rating:** 5
**Confidence:** 5

**Summary:**

This paper introduces a novel method, SSL-GM, for graph inference acceleration. The primary idea is to bridge GNNs and Multi-Layer Perceptrons (MLPs) through Self-Supervised Learning (SSL) to integrate structural information into MLPs. SSL-GM employs self-supervised contrastive learning to align the representations of GNNs and MLPs. Besides, it uses non-parametric aggregation, augmentation, and reconstruction regulation to avoid potential model collapse, improve model training, and prevent representation shift, respectively. The extensive experiments demonstrate SSL-GM's empirical superiority over existing models in terms of both efficiency and effectiveness.

**Strengths:**

* It introduces a novel approach, SSL-GM, which integrates structural information into MLPs by contrastive learning between GNN and MLP outputs.
* The empirical results over 10 datasets demonstrate the effectiveness of SSL-GM.
* The paper is well-written and structured, with clear explanations of the methodology and theoretical insights.

**Weaknesses:**

* In this paper, the structure-aware ability of SSL-GM is supported by the objective $\mathcal{L}_{cont}$. However, its success depends on the quality of the learned representations from the GNN, rather than solely on the capabilities of the student MLP. In other words, $\mathcal{L}{cont}$ enforces the student MLP to output representations similar to the aggregated representations from the GNN but does not inherently empower the student with structure-aware abilities. For unseen nodes during training, the student MLP may encounter challenges in learning structure-aware representations without the supervision provided by the GNN.
* It's worth considering whether addressing the representation shift issue is necessary. The absence of the reconstruction strategy only leads to a slight decrease in performance (as shown in Table 1, "w/o Rec." column), indicating that this strategy may not be indispensable in some cases.
* To enhance reproducibility and ensure accurate results, it would be helpful if the authors could provide detailed information about the experimental environment used for SSL-GM.  I used the Google Colab platform to rerun the source code but obtained $83.80_{\pm0.46}$ in the Cora dataset compared to  $84.60_{\pm0.24}$ in the paper.

**Questions:**

Please refer to the Weaknesses section.

---

> ### Author Response · Authors · 2023-11-15
> **Response to Reviewer rTyj - Part 1**
>
> Thank you for your detailed comments and constructive suggestions. To address your concerns, we provide the following responses.
>
> > Q1. In this paper, the structure-aware ability of SSL-GM is supported by the objective $L_{cont}$. However, its success depends on the quality of the learned representations from the GNN, rather than solely on the capabilities of the student MLP. In other words, $L_{cont}$ enforces the student MLP to output representations similar to the aggregated representations from the GNN but does not inherently empower the student with structure-aware abilities. For unseen nodes during training, the student MLP may encounter challenges in learning structure-aware representations without the supervision provided by the GNN.
>
> Thank you for your insightful comment regarding the structure-aware capabilities of our SSL-GM model. We appreciate the opportunity to clarify how our model addresses the limitations of traditional GNN-MLP methods and its independence from GNN supervision.
>
> **Addressing Limitations of GNN-MLP Methods.** Our primary goal with SSL-GM is to overcome the challenges faced by existing GNN-MLP models that depend heavily on knowledge distillation from teacher GNNs to student MLPs. We found that the performance of student MLPs in these models is tightly bound to the quality of pre-trained GNNs, leading to limited generalizability, especially for unseen nodes. traditional methods, SSL-GM injects high-order information into MLPs through self-supervised learning (SSL), negating the need for a well pre-trained GNN. This approach ensures that the GNN representations serve as high-order approximations of the MLP representations.
>
> **Inherent High-Order Information in MLP Representations.** By maximizing the mutual information between MLP representations and the approximated GNN representations, we enable the MLP to inherently contain high-order structural information. This mechanism is key to ensuring the structure-aware capabilities of the student MLP independently of the GNN. To validate this capability, we conducted experiments in inductive and cold-start settings, assessing the model's effectiveness on unseen nodes. The results, as presented in Tables 2 and 3 of our paper, demonstrate the robustness of SSL-GM in these challenging scenarios.
>
> **Inductive and Cold-Start Settings.** The inductive and cold-start experiments provide concrete evidence of SSL-GM's ability to generalize well to unseen nodes, a critical measure of its structure-aware capabilities. These settings were specifically chosen to test the model performance in scenarios where traditional GNN-MLP methods typically struggle.
>
>
> > Q2. It's worth considering whether addressing the representation shift issue is necessary. The absence of the reconstruction strategy only leads to a slight decrease in performance (as shown in Table 1, "w/o Rec." column), indicating that this strategy may not be indispensable in some cases.
>
> Thank you for your observation regarding the impact of the reconstruction strategy on our SSL-GM model's performance. We appreciate this opportunity to discuss the conditions under which this strategy is particularly impactful.
>
> We concur that the significance of the representation shift issue is closely tied to the size and complexity of the graph. In large-scale graphs with extensive and complex structures, addressing representation shift becomes increasingly important. As illustrated in our experiments, particularly on datasets like Arxiv, the absence of the reconstruction regularizer led to a noticeable performance drop (nearly 1 percentage point). This indicates that for large-scale graphs, the reconstruction strategy plays a crucial role in maintaining the integrity and quality of representations.
>
> Our findings highlight the need for a strategic use of the reconstruction strategy in SSL-GM, applying it judiciously based on the characteristics of the graph at hand. For larger graphs, the strategy is indispensable for maintaining high-quality representations and overall performance. We plan to continue exploring the dynamics of representation shift across different graph sizes and types, further refining our understanding of when and how to effectively implement reconstruction strategies in our model.

---

> > ### Comment · Reviewer_rTyJ · 2023-11-20
> > **Questions about Inductive Setting and Cold-start Setting.**
> >
> > Thank you for the clarification. I appreciate the authors' responses in helping me understand the work better. However, I still have concerns regarding the experimental outcomes presented in Table 2 and Table 3, which showcase the primary contribution of SSL-GM in terms of experiments.
> >
> > If I understand correctly, the cold-start setting can be seen as a non-structure-in-inductive-set version of the inductive setting. In the case of NOSMOG, which relies on positional encoding, it is expected to experience performance drops compared to the inductive setting, as observed in Table 3 and Table 2. For MLP and SSL-GM, which do not require structural information from the inductive set, slight performance differences between the two settings can be attributed to variations in splits or seeds.
> >
> > However, the performance discrepancy in the cold-start setting for GLNN, which also disregards structural information, is quite significant. For instance, on Cora, the performance is 78.34 in the inductive setting compared to 71.96 in the cold-start setting. Similarly, on Amazon-CS/Amazon-Photo/Arxiv, the performance is 87.04/93.28/63.53 in the inductive setting compared to 83.98/91.05/60.55 in the cold-start setting. It is expected that the GNN teacher should perform similarly in the observed set, which is the same or similar for both settings, indicating comparable inference abilities acquired by GLNN. Regarding Citeseer, Co-CS, and Co-Phy, the results appear to be reasonable.
> >
> > These variations in both settings suggest that there might be significant differences when evaluating GLNN in the two settings. It raises concerns about the experimental results of the baselines, as it seems to be an unfair comparison. Additionally, the details of the cold-start setting, including implementation and hyperparameters, are not provided in the source codes, which adds further uncertainty.

---

> > > ### Author Response · Authors · 2023-11-20
> > > **Response to the Questions about Settings.**
> > >
> > > Thank you for your queries regarding the experimental settings used in our study. We understand the importance of distinguishing between various settings and their implications. **It is crucial to emphasize that the cold-start setting in our paper is not synonymous with the non-structure-in-inductive-set version.** The inductive (production) setting can be viewed as an interpolation between transductive and inductive settings. This hybrid approach allows us to capture the nuances and complexities of real-world applications more effectively. We explain the difference between settings following:
> > >
> > > - **Transductive Setting.** We consider a graph $G = (V, E)$ in which all nodes are visible during the training stage. We partition the nodes into three non-overlapping sets: $V = V_{train} \sqcup V_{val} \sqcup V_{test}$. In the supervised setting, we train the encoder using $V_{train}$ and evaluate its performance on $V_{val}$ and $V_{test}$. In the self-supervised setting, we train the encoder on $V$ and utilize $V_{trans}$ for training the downstream head, while using $V_{val}$ and $V_{test}$ for evaluation. We perform the split 10 times to evaluate the quality of the learned representations to alleviate the impact of randomness.
> > >
> > > - **Inductive (Production) Setting.** In the production setting, we partition a graph $G = (V, E)$ into transductive set $G^T$ and inductive set $G^I$, where $G = G^{T} \sqcup G^{I}$ and $\emptyset = G^{T} \sqcap G^{I}$. $G^{T} = (V^{T}, E^{T})$, containing 80\% of the nodes, is used for training, while $G^{I} = (V^{I}, E^{I})$, containing the remaining 20\% of the nodes, remains unseen during training. We further partition the nodes in $G^{T}$ into non-overlapping training, validation, and testing sets, denoted as $V^T = V^T_{train} \sqcup V^T_{valid} \sqcup V^T_{test}$. The model is trained on $G^{T}$, and we report the transductive results on $V^{T}_{test}$ and inductive results on $V^{I}$. **We interpolate the results of these two settings to represent production results.** We treat nodes in the inductive set as disconnected from nodes in the transductive set, even during inference, creating a more challenging out-of-distribution setting.
> > >
> > > - **Cold-start Setting.** The cold-start setting is closely similar to inductive (production) setting, but we assume the nodes in the inductive set are isolated, $G^{C} = {( V^{I}, \emptyset )}$. This is a more challenging yet practical setting, as new agents often emerge independently in real-world systems. We follow the production setting to train the model on transductive set $G^T$ but **only report the model performance on cold-start set $G^C$ to assess cold-start performance**. For models that rely on positional encoding, such as GENN and NOSMOG, we set the positional embeddings as zero vectors.
> > >
> > > ---
> > >
> > > We also present a comprehensive table of production settings, including transductive, inductive, and production results in Table 13 in the paper. For GLNN that do not use any structural information in inference phase, you can find a rigorous matching between the results presented in Table 3 and the results in Table 13 under *ind*.

---

> ### Author Response · Authors · 2023-11-15
> **Response to Reviewer rTyj - Part 2**
>
> > Q3. To enhance reproducibility and ensure accurate results, it would be helpful if the authors could provide detailed information about the experimental environment used for SSL-GM.
>
> Thank you for emphasizing the importance of reproducibility in our research. We agree that providing detailed information about our experimental setup is crucial for validating and replicating our results. Below is a comprehensive outline of our experimental environment:
> ```
> python==3.8.0
> torch==2.10
> torch_geometric==2.3
> torch_scatter==2.1.2
> torch_cluster==1.6.3
> torch_sparse==0.6.18
> ogb==1.3.6
> ```
>
>
> **Code Availability and Bug Mitigation.** We have meticulously reviewed and updated our code to address potential bugs and ensure consistent performance. The updated code is readily available for download and can be executed in environments such as Google Colab.
>
> **Step-by-Step Guide for Replication in Colab.**
>
> - **Code Setup.** Download and upload the updated code to Google Colab.
> - **Environment Installation.** Execute the following commands in Colab to set up the necessary environment.
>
> ```
> !pip3 install torch torchvision torchaudio --index-url https://download.pytorch.org/whl/cu118
> !pip install torch_geometric
> !pip install pyg_lib torch_scatter torch_sparse torch_cluster torch_spline_conv -f https://data.pyg.org/whl/torch-2.1.0+cu118.html
> !pip install ogb
> ```
>
> - **Execution of Experiments.** Run the code using the command.
>
> ```
> !python main.py --setting trans --dataset cora --use_params --seed 1
> ```
>
> For comprehensive results, average the outcomes over ten different seeds (1 to 10), and make 10 train/validation/test splits for each seed to minimize the impact of randomness. In our experiments on Colab, the accuracy for Cora over 10 seeds typically peaks at epochs 200 or 300. The averaged accuracy is $84.63±0.36$.
>
> **Ensuring Reproducibility.**
>
> We encourage researchers to follow these guidelines for replicating our experiments. Our approach ensures a high degree of reproducibility and accuracy, allowing others to validate and build upon our work confidently.
>
> We hope this detailed information assists in the replication and verification of our experimental results. Our commitment to open and reproducible research is paramount, and we are grateful for the opportunity to share these details.
>
> ---
>
> In light of these responses, we hope we have addressed your concerns, and hope you will consider increasing your score. If we have left any notable points of concern unaddressed, please do share and we will attend to these points.

---

> ### Comment · Reviewer_rTyJ · 2023-11-21
> **Question about the implications of SSL-GM in practical applications**
>
> Thank you for the authors' response. I appreciate their clarification, which has addressed my concerns regarding the experimental settings.
>
> I would like to further discuss the implications of SSL-GM in practical applications. While SSL-GM offers improvements in addressing the limitations of GNN-MLP methods, it introduces **an additional training process** using a Logistic regression function for test nodes or graphs based on downstream tasks. This approach may hinder the inherited advantage of GNN-MLP techniques, particularly in latency-constrained applications. For example, in node classification tasks where the deployed model needs to handle an arbitrary number of test nodes at any given time, traditional GNN-MLP methods can perform inference on all the nodes simultaneously, while SSL-GM requires additional training time, **resulting in increased latency**. This trade-off between performance and speed could potentially reduce the primary contribution of SSL-GM in addressing the limitations of GNN-MLP methods.
>
> Additionally, I have noticed the extension of SSL-GM to supervised learning. It is worth mentioning that the performance of SSL-GM drops significantly on the Cora dataset in this extension. Furthermore, the reported results are limited to the semi-supervised setting. I am particularly interested in the evaluation of SSL-GM in the inductive and cold-start settings, which would help validate its value in addressing the limitations of GNN-MLP methods while preserving their practical advantages.

---

### Official Review · Reviewer_eCUd · 2023-10-24

**Soundness:** 2 fair
**Presentation:** 3 good
**Contribution:** 2 fair
**Rating:** 3
**Confidence:** 4

**Summary:**

This paper considers the problem of accelerating GNN inference by training an MLP on the node features. It proposes to use self-supervised learning to train the MLP and achieves strong empirical results.

**Strengths:**

1.	The paper is well-written and easy to follow.
2.	The empirical results are rich and significantly outperform the baselines.

**Weaknesses:**

1.	Theorem 1 is possibly wrong. I check the proof in the appendix, which basically connects each term in (17) to each term in (13) and leads to two problems. (1) The authors seem that minimizing each term in (17) also minimizes the corresponding term in (13). Pls prove these conclusions rigorously instead of only using intuitively explanations. (2) Even if the first point holds, (13) minimizes the summarization of 4 terms, and thus the minimizer may not be the minimizers of the 4 terms. Thus, there is no guarantee that (13) and (16) will have the same minimizer. Nowadays, many machine learning papers have theorems but my opinion is that theorems should be rigorous. Moreover, I failed to follow the mutual information part of Section 5, especially how the authors transform different models into mutual information forms. If this can be done, pls conduct the mathematical transformations in rigorous ways.
2.	It is unclear what are the challenges of using self-supervised learning (SSL) to train MLP and what are the new designs of the paper. SSL is widely used for graph learning, and the author should be very specific in the challenges of using it to train MLP. Currently, descriptions of the weak points of existing works, e.g., “insufficient exploration of structural information when inferring unseen nodes”, “cannot fully model the structural information” are rather vague. In section 3, the authors propose several techniques and loss terms, e.g., alignment loss, data augmentation, and reconstruction loss. These are not new for SSL, which are also evidenced by the citations provided by the authors. The question is that what are the new things proposed by the authors. The paper will be stronger if the authors can connect the challenges and the proposed new techniques.
3.	Experiments can be improved. (1) Pls provide the time for model training, which is also an important practical consideration. (2) Pls run the experiments on large practical datasets, e.g., Papers100M, MAG240M, and IGB.
====================================================
I have reade the author response. Instead of addressing my concerns, the author response raises more concerns. As such, I decide to lower my rating.

**Questions:**

NA

---

> ### Author Response · Authors · 2023-11-15
> **Response to Reviewer eCUd - Part 1**
>
> Thank you for your detailed comments and constructive suggestions. To address your concerns, we provide the following responses.
>
> > Q1. The authors seem that minimizing each term in (17) also minimizes the corresponding term in (13). Pls prove these conclusions rigorously instead of only using intuitively explanations.
>
> Thank you for highlighting the need for a more rigorous explanation of the connections between the terms in equations (17) and (13). We appreciate the opportunity to delve deeper into the mathematical underpinnings of our model. First of all, we want to emphasize that we consider Eq. (13) only as an instantiation of Eq. (17). Here we will reinterpret the proof process.
>
> **Information Bottleneck Perspective.** We start by reinterpreting equation (17) from the perspective of the information bottleneck principle. The equation aims to jointly optimize four terms, each representing a different aspect of the information bottleneck:
>
> $$
> \mathbf{T}^* = argmin_{\mathbf{H}, \mathbf{Z}}  \lambda H(\mathbf{H} | \mathcal{G}_\mathcal{I})  + \lambda H(\mathbf{Z} | \mathbf{H}, {\mathcal{G}}_I) + H(\mathbf{Z}) + H(\mathbf{H} | \mathbf{Z})
> $$
>
> where $\mathbf{T} = (\mathbf{H}, \mathbf{Z})$, $\mathbf{H}$ and $\mathbf{Z}$ are the representations learned by MLPs and GNNs, respectively.
>
> **Modeling the Terms.** We model the first term by minimizing the distance between $\mathbf{H}$ and $\mathbf{F}$, represented as $\| \mathbf{H} - \mathbf{F}^* \|$. For the second term, we minimize the distance between $\mathbf{F}$ and $\mathbf{Z}$, corresponding to $\| \mathbf{Z} - \mathbf{F}^* \|$. The third term, $H(\mathbf{H} | \mathbf{Z})$, is approached by optimizing the covariance between $\mathbf{Z}$ and $\mathbf{H}$, expressed as $\sum_i \text{Cov}(\mathbf{H}_i, \mathbf{Z}_i)$. To minimize the last term, $H(\mathbf{Z})$, we make $\mathbf{Z}$ invariant to a specific variable, using $\mathbf{Z}$ to reconstruct $\mathbf{X}$.
>
> Combining these four terms, with some transformation, we derive our final optimization objective:
> $$
> \mathbb{E}_{\hat{\mathbf{A}}, \hat{\mathbf{X}}} \left[ \| \hat{\mathbf{H}} - \mathbf{F}^* \|^2 + \| \hat{\mathbf{Z}} - \mathbf{F}^* \|^2 +  \| \mathcal{D}(\hat{\mathbf{Z}}) - \mathbf{X} \|^2  | \hat{\mathbf{A}}, \hat{\mathbf{X}}  \right] - 2  {\mathbb{E}} \left[ \sum_i \text{Cov}(\hat{\mathbf{H}}_i, \hat{\mathbf{Z}}_i) |  \hat{\mathbf{A}}, \hat{\mathbf{X}}, \mathbf{F}^* \right]
> $$
>
> **Clarification on the Connection Between Eq. (13) and Eq. (17).** It is important to note that a direct, rigorous connection between the terms in equations (17) and (13) is not necessary. Our final objective serves as a practical implementation of the information bottleneck concept, tailored to the specific context of our model.
>
> In summary, our approach to optimizing equation (17) is grounded in the principles of the information bottleneck, with each term contributing to the overarching goal of learning robust and informative representations. We believe this explanation provides a more rigorous understanding of our method's mathematical foundation.
>
>
> > Q2. Even if the first point holds, (13) minimizes the summarization of 4 terms, and thus the minimizer may not be the minimizers of the 4 terms. Thus, there is no guarantee that (13) and (17) will have the same minimizer. Nowadays, many machine learning papers have theorems but my opinion is that theorems should be rigorous.
>
> Thank you for your critical observation regarding the relationship between the minimizers of equations (13) and (17). We appreciate this opportunity to clarify our approach and address the need for theoretical rigor.
>
> **Differentiation Between Eq. (13) and (17).** As discussed, equation (13) represents an implementation of the broader conceptual framework outlined in equation (17). Our approach to minimizing equation (17) through equation (13) is a practical instantiation rather than a direct equivalence. We acknowledge that the minimizer of equation (13) may not necessarily be the same as the individual minimizers of the four terms in equation (17). The composite nature of equation (13) means that its minimization involves a balance among these terms, which may not align perfectly with their individual minimization.
>
> **Future Research Directions.** This realization forms the basis for our future research. We aim to develop more refined implementations that bring the minimization of equation (13) closer to the ideal minimization of equation (17). This will involve exploring methods with lower upper bounds and potentially tighter convergence to the theoretical optimum. We also recognize the importance of theoretical rigor in our assertions and theorems. Moving forward, we will focus on strengthening the mathematical foundations of our model, ensuring that our theoretical claims are robustly supported by rigorous analysis.

---

> > ### Comment · Reviewer_eCUd · 2023-11-22
> > **Confused by the response**
> >
> > (1) In the Information Bottleneck Perspective part, you directly translate some entropy expressions into some norm expressions. In my opinion, this only holds when you assume that data (or anything that can be assumed) follow certain distribution. However, the assumption part is missing and the derivation is entirely intuitively.
> >
> > (2) Clarification on the Connection Between Eq. (13) and Eq. (17). You say that a rigorous connection is not required. However, according to my understanding, in a rigorous proof, every step should be connected in reasonable ways. I do not understand this part.
> >
> > If you are actually proving Eq. (17), pls state the theorem in terms of Eq. (17) rather than a stronger statement in  Eq. (13).

---

> ### Author Response · Authors · 2023-11-15
> **Response to Reviewer eCUd - Part 2**
>
> **Remark.** The current implementation, as outlined in equation (13), represents the first step in an ongoing process to optimize equation (17). We are committed to continuous improvement and refinement of our methods. Your feedback is invaluable in guiding our efforts towards more rigorous and effective model development. It helps us identify areas for theoretical and practical enhancement in our research.
>
> > Q3. Moreover, I failed to follow the mutual information part of Section 5, especially how the authors transform different models into mutual information forms. If this can be done, pls conduct the mathematical transformations in rigorous ways.
>
> Thank you for pointing out the need for a more rigorous and detailed mathematical explanation of the transformations into mutual information forms for various models. We appreciate this opportunity to delve into the specifics of these transformations.
>
> **Transformation Process.** Based on the findings in [1], optimizing the cross-entropy $\text{CE}(\mathbf{X}, \mathbf{Y})$ between $\mathbf{X}$ and $\mathbf{Y}$ corresponds to maximizing the mutual information $\text{MI}(\mathbf{X}, \mathbf{Y}) = \sum_{i\in \mathcal{V}} I(\mathbf{y}_i, \mathbf{x}_i)$ between $\mathbf{X}$ and $\mathbf{Y}$.
>
> [1] Malik Boudiaf, Jerome Rony, Imtiaz Masud Ziko, Eric Granger, Marco Pedersoli, Pablo Piantanida, and Ismail Ben Ayed. A unifying mutual information view of metric learning: cross-entropy vs. pairwise losses. In ECCV, 2020.
>
> **Model-Specific Transformation.** The specific transformation is summarized in the following table.
>
> |               | Objective                                                                      |
> |---------------|------------------------------------------------------------------------------------|
> | MLP           | $\sum_{i \in \mathcal{V}} I(\mathbf{y}_i, \mathbf{x}_i)$                           |
> | GNN           | $\sum_{i \in \mathcal{V}} I(\mathbf{y}_i; \mathcal{S}_i)$                          |
> | GLNN          | $\sum_{i \in \mathcal{V}} I(\mathbf{x}_i; \mathbf{y}_i \| \mathcal{S}_i)$           |
> | NOSMOG & GENN | $\sum_{i \in \mathcal{V}} I(\mathbf{x}_i; \mathbf{y}_i \| \mathcal{S}_i) + I(\mathbf{y}_i; \mathbf{A}^{[i]})$    |
> | SSL-GM (ours) | $\sum_{i \in \mathcal{V}} I(\mathbf{y}_i; \mathbf{x}_i \| \mathcal{S}_i) + I(\mathbf{x}_i; \mathcal{S}_i)$ |
>
> - **MLPs.** For MLPs optimizing the cross-entropy between node features and labels, the objective transforms directly into $\sum_{i\in \mathcal{V}} I(\mathbf{y}_i, \mathbf{x}_i)$.
>
> - **GNNs.** GNNs consider both structural information and node features. We define the computational graph surrounding node $i$ as $S_i = (\mathbf{X}^{[i]}, \mathbf{A}^{[i]})$, where $\mathbf{X}^{[i]}$ is the node features of neighborhoods of target node $i$ and $\mathbf{A}^{[i]}$ is the corresponding subgraph structure, transforming the GNN learning objective into $\sum_{i \in \mathcal{V}} I(\mathbf{y}_i; \mathcal{S}_i)$.
>
> - **GLNN.** For GLNN that uses MLPs to mimic the predictions of GNNs, we denote the outputs of GNNs as $y_i | S_i$ and thus convert the learning objective $\sum_{i \in \mathcal{V}} I(\mathbf{x}_i; \mathbf{y}_i | \mathcal{S}_i)$.
>
> - **NOSMOG and GENN.** The following works of GLNN further add structural information in the learning objective, denoted as $\sum_{i \in \mathcal{V}} I(\mathbf{x}_i; \mathbf{y}_i | \mathcal{S}_i) + I(\mathbf{y}_i; \mathbf{A}^{[i]})$.
>
> - **SSL-GM (Ours).** Our proposed SSL-GM applies contrastive learning to maximize the mutual information between the node features and the high-order representations first and then use the learned representations to do classification, denoted as $\sum_{i \in \mathcal{V}} I(\mathbf{y}_i; \mathbf{x}_i | \mathcal{S}_i) + I(\mathbf{x}_i; \mathcal{S}_i)$.
>
> **Detailed Explanation.** In the context of MLPs, the focus is on maximizing the mutual information solely between the labels and node features. This approach, however, does not account for structural information inherent in the graph data, potentially limiting the depth of understanding that can be derived from the features alone. GNNs extend this concept by maximizing the mutual information between the labels and the corresponding subgraphs. This approach captures both semantic (feature-based) and structural (graph topology) information, offering a more holistic representation of the graph data. GLNNs further enhance this framework by maximizing the mutual information between node features and the soft labels generated by pre-trained GNNs. This approach leverages high-level abstract information retained in GNN outputs, enriching the feature representation with insights gleaned from the graph structure. NOSMOG and GENN incorporate an additional layer of complexity by maximizing mutual information between the subgraph of the target node and its labels. This addition emphasizes fine-grained structural information, further enriching the model understanding of the graph's intricacies.

---

> ### Author Response · Authors · 2023-11-15
> **Response to Reviewer eCUd - Part 3**
>
> In our SSL-GM approach, the second term of mutual information maximization is directly optimized. We posit that when this second term is maximized, it effectively transforms the first term into $\sum_{i \in \mathcal{V}} I(y_i; x_i | S_i) = \sum_{i \in \mathcal{V}} I(y_i; S_i)$. This transformation aligns our objective with that of traditional GNNs, focusing on maximizing the mutual information between labels and subgraphs, thereby capturing both semantic and structural elements of the graph.
>
> In summary, each model adopts a unique approach to maximizing mutual information, progressively incorporating more complex layers of graph information. Our SSL-GM method synthesizes these approaches, aiming to capture a comprehensive representation of both node features and graph structure.
>
> > Q4. It is unclear what are the challenges of using self-supervised learning (SSL) to train MLP and what are the new designs of the paper. SSL is widely used for graph learning, and the author should be very specific in the challenges of using it to train MLP. Currently, descriptions of the weak points of existing works, e.g., “insufficient exploration of structural information when inferring unseen nodes”, “cannot fully model the structural information” are rather vague. In section 3, the authors propose several techniques and loss terms, e.g., alignment loss, data augmentation, and reconstruction loss. These are not new for SSL, which are also evidenced by the citations provided by the authors. The question is that what are the new things proposed by the authors. The paper will be stronger if the authors can connect the challenges and the proposed new techniques.
>
>
> Thank you for your insightful comment. We appreciate the opportunity to clarify the innovative aspects of our work, particularly in addressing the limitations of existing GNN-MLP methods in inductive, cold-start, and label sparsity settings.
>
>
> - **Insight on Generalization Issues.** Our core contribution does not lie in new model architectures and new objectives. Instead, our work identifies the lack of generalization in current methods as a key issue, stemming from the limitations of knowledge distillation (KD) that aligns GNN and MLP outputs only at the label level. We propose that aligning these models in the representation space using self-supervised learning (SSL) can yield more generalizable representations. This insight represents a significant shift from existing practices.
>
> - **Challenges in SSL Alignment.** Different from KD methods, aligning GNNs and MLPs under SSL is challenging due to the absence of a pre-trained GNN. To this end, we convert the objective to jointly train GNN and MLP by aligning the 0-hop representations (MLP output) with high-order representations (GNN output). Our approach is applicable with various objectives like Bootstrap, InfoNCE, or MSE. This flexibility in alignment strategy is a novel contribution to the field.
>
> - **Addressing Representation Inconsistency and Model Collapse.** We observed inconsistency in representations between GNNs and MLPs, potentially leading to model collapse (Appendix D.3). Our solution involves using MLP outputs to approximate GNN outputs, a strategy akin to SGC but applied to a different problem context.
>
> - **Graph Augmentation and Reconstruction Regularizer.** We further incorporate data augmentation to enhance the quality of the learned representations. However, given that structural permutation in augmentation can significantly shift GNN representations, we introduce a reconstruction regularizer to mitigate representation shift. This addition is critical in ensuring the stability and reliability of the learned representations.
>
>
> > Q5. Pls provide the time for model training, which is also an important practical consideration.
>
> Thank you for your valuable feedback. In response to your comment, we have presented additional experimental results in Appendix C of our paper to further validate the efficacy of our SSL-GM model in inference acceleration.
>
> - **Accuracy vs Efficiency.** We present a comprehensive comparison between our method and other acceleration methods. The results showcase the inference time and accuracy for various methods, including SGC, APPNP, Quantization (QSAGE), Pruning (PSAGE), and Neighbor Sample on Flickr and OGB-Arxiv. We have intentionally omitted GNN-MLP methods that were compared in Figure 3 for brevity. Please note that the time consumption of SAGE and BGRL are the same since the learning process of SSL does not affect inference.
>
> - **Different Settings.** To ensure a thorough evaluation, we chose three datasets across all node classification settings, including transductive, inductive, and cold-start. Our results, as shown in the additional tables, clearly indicate that SSL-GM outperforms existing methods in terms of efficiency and accuracy.
>
> [Experimental results are following]

---

> > ### Comment · Reviewer_eCUd · 2023-11-22
> >
> > I think you are reporting inference time in the tables below. However, I think training time should be reported for readers to understand the cost.

---

> ### Author Response · Authors · 2023-11-15
> **Response to Reviewer eCUd - Part 4**
>
> | Datasets |  | SAGE      | BGRL  | SGC                  | APPNP                | QSAGE                | PSAGE                | Neighbor Sample      | **SSL-GM** |
> |------------|------------------|-----------|-------|----------------------|----------------------|----------------------|----------------------|----------------------|----------------------------|
> | Flickr     | Time (ms) | 80.7  | 80.7 (1.00$\times$)  | 76.9 (1.05$\times$)  | 78.1 (1.03$\times$)  | 70.6 (1.14$\times$)  | 67.4 (1.20$\times$)  | 25.5 (3.16$\times$)        | **0.9 (89.67$\times$)** |
> |                              | Acc (\%)  | 47.17 | 49.12                | 47.35                | 47.53                | 47.22                | 47.25                | 47.01                      | **49.27**                |
> | Arxiv       | Time (ms) | 314.7 | 314.7 (1.00$\times$) | 265.9 (1.18$\times$) | 284.1 (1.11$\times$) | 289.5 (1.09$\times$) | 297.5 (1.06$\times$) | 78.3 (4.02$\times$)        | **2.5 (125.88$\times$)** |
> |                              | Acc (\%)  | 68.52 | 69.29                | 68.93                | 69.10                | 68.48                | 68.55                | 68.35                      | **70.23**                |
>
>
>
> |           |        | Trans      | Trans          | Ind        | Ind            | Cold-start | Cold-start     |
> |-----------|--------|------------|----------------|------------|----------------|------------|----------------|
> | Dataset   | Models | Time (ms)  | Acc (\%)       | Time (ms)  | Acc (\%)       | Time (ms)  | Acc (\%)       |
> | Pubmed    | SAGE   | 73         | 85.94          | 15         | 85.04          | 15         | 77.98          |
> |           | SGC    | 64         | 85.28          | 14         | 85.22          | 13         | 76.10          |
> |           | NOSMOG | 5          | 86.18          | 3          | 83.84          | 3          | 81.48          |
> |           | SSL-GM | **3** | **86.99** | **3** | **86.47** | **3** | **86.44** |
> | Amazon-CS | SAGE   | 103        | 88.88          | 31         | 87.24          | 25         | 61.01          |
> |           | SGC    | 89         | **89.31** | 26         | 87.12          | 24         | 63.08          |
> |           | NOSMOG | 5          | 87.64          | 4          | 86.61          | 4          | 81.95          |
> |           | SSL-GM | **4** | 88.46          | **3** | **87.65** | **3** | **87.58** |
> | Arxiv     | SAGE   | 485        | **72.05** | 315        | 68.52          | 305        | 43.47          |
> |           | SGC    | 410        | 69.95          | 266        | 68.93          | 250        | 42.08          |
> |           | NOSMOG | 6          | 70.84          | 4          | 69.10          | 4          | 61.64          |
> |           | SSL-GM | **4** | 71.12          | **3** | **70.23** | **3** | **66.13** |
>
>
>
> > Q7. Pls run the experiments on large practical datasets, e.g., Papers100M, MAG240M, and IGB.
>
> Thank you for your valuable suggestion to extend our experiments to include large-scale practical datasets like Papers100M, MAG240M, and IGB. Although these datasets are seldomly used in graph learning experiments, we appreciate your emphasis on the importance of testing our model in diverse and challenging real-world scenarios.
>
> We fully acknowledge the significance of evaluating our model on these large datasets. Such an expansion in our experimental scope would undoubtedly provide deeper insights into the scalability and robustness of our model. We are committed to incorporating this suggestion into our future research plans. Conducting experiments on these larger datasets will be a key focus in our subsequent work, allowing us to further validate and refine our model.
>
> ---
> In light of these responses, we hope we have addressed your concerns, and hope you will consider increasing your score. If we have left any notable points of concern unaddressed, please do share and we will attend to these points.

---

> > ### Comment · Reviewer_eCUd · 2023-11-22
> >
> > I cannot agree with the statement that "these datasets are seldomly used in graph learning experiments". A simple search can easily find a dozen graph learning papers that use these datasets.

---

### Official Review · Reviewer_WTGk · 2023-10-25

**Soundness:** 3 good
**Presentation:** 3 good
**Contribution:** 3 good
**Rating:** 6
**Confidence:** 2

**Summary:**

This paper studies an important problem: how to accelerate graph inference and improve model generalization. For this paper, the authors propose SSL-GM to bridge graph context-aware GNNs and neighborhood dependency-free MLPs with SSL. In addition, the authors also provide theoretical analysis to prove the generalization capability of SSL-GM. Furthermore, the extensive experimental results show that the solution mentioned in this paper not only accelerates GNN inference but also exhibits significant performance improvements over vanilla MLPs.

**Strengths:**

1. The problem studied in this paper is fundamental in the graph neural network area.
2. The solution mentioned in this paper is basic and the performance of the method mentioned in this paper is great. The experimental results are extensive.
3. This paper develops a theoretical analysis.
4. The presentation of this paper is so clear that I can follow the paper easily.

**Weaknesses:**

1. The experimental results only contain the performance over node classification and graph classification. Is it possible to evaluate the proposed method over link prediction?
2. It seems that this paper only borrows some ideas from contrastive learning. Based on contrastive learning, this paper develops a new objective function that can be used to solve the model generalization problem. Therefore, could the authors highlight some contributions here? I think it is a good paper but the contribution of this paper is a little bit marginal. If the authors are able to emphasize their contributions here, I am willing to improve my rate.

**Questions:**

See Strengths and Weaknesses.

---

> ### Author Response · Authors · 2023-11-15
> **Response to Reviewer WTGk**
>
> Thank you for your detailed comments and constructive suggestions. To address your concerns, we provide the following responses.
>
> > Q1. The experimental results only contain the performance over node classification and graph classification. Is it possible to evaluate the proposed method over link prediction?
>
> Thank you for your insightful question regarding the application of our model to link prediction tasks. We appreciate this opportunity to discuss the adaptability of our method to different graph analysis tasks.
>
> **Applying the Model to Link Prediction.** Our model adopts self-supervised learning (SSL) approach that is inherently flexible and can be applied to link prediction without requiring significant modifications to the graph training process. To represent links, we suggest concatenating the corresponding node representations. This concatenated representation will then serve as the input to the logistic regression model for predicting the presence or absence of a link. For the inference phase in link prediction, we propose adopting a linear protocol where the parameters of the encoder are frozen. This means that during inference, only the logistic regression model, which is used for predicting links, will be trained. The link prediction task can be approached in a standard manner, involving the sampling of some edges from the original graph as positive examples and performing negative sampling for negative edges.
>
> **Future Work.** While our current experimental results do not include link prediction, we acknowledge the importance of this task in graph analysis. Therefore, we are considering extending our experimental evaluation to include link prediction, providing a comprehensive view of the applicability across various graph-related tasks.
>
> > Q2. It seems that this paper only borrows some ideas from contrastive learning. Based on contrastive learning, this paper develops a new objective function that can be used to solve the model generalization problem. Therefore, could the authors highlight some contributions here? I think it is a good paper but the contribution of this paper is a little bit marginal. If the authors are able to emphasize their contributions here, I am willing to improve my rate.
>
>
> Thank you for your insightful comment. We appreciate the opportunity to clarify the innovative aspects of our work, particularly in addressing the limitations of existing GNN-MLP methods in inductive, cold-start, and label sparsity settings.
>
> - **Insight on Generalization Issues.** Our core contribution does not lie in new model architectures and new objectives. Instead, our work identifies the lack of generalization in current methods as a key issue, stemming from the limitations of knowledge distillation (KD) that aligns GNN and MLP outputs only at the label level. We propose that aligning these models in the representation space using self-supervised learning (SSL) can yield more generalizable representations. This insight represents a significant shift from existing practices.
>
> - **Challenges in SSL Alignment.** Different from KD methods, aligning GNNs and MLPs under SSL is challenging due to the absence of a pre-trained GNN. To this end, we convert the objective to jointly train GNN and MLP by aligning the 0-hop representations (MLP output) with high-order representations (GNN output). Our approach is applicable with various objectives like Bootstrap, InfoNCE, or MSE. This flexibility in alignment strategy is a novel contribution to the field.
>
> - **Addressing Representation Inconsistency and Model Collapse.** We observed inconsistency in representations between GNNs and MLPs, potentially leading to model collapse (Appendix D.3). Our solution involves using MLP outputs to approximate GNN outputs, a strategy akin to SGC but applied to a different problem context.
>
> - **Graph Augmentation and Reconstruction Regularizer.** We further incorporate data augmentation to enhance the quality of the learned representations. However, given that structural permutation in augmentation can significantly shift GNN representations, we introduce a reconstruction regularizer to mitigate representation shift. This addition is critical in ensuring the stability and reliability of the learned representations.
>
> ---
>
> In light of these responses, we hope we have addressed your concerns, and hope you will consider increasing your score. If we have left any notable points of concern unaddressed, please do share and we will attend to these points.

---

### Official Review · Reviewer_N5QN · 2023-10-31

**Soundness:** 2 fair
**Presentation:** 2 fair
**Contribution:** 2 fair
**Rating:** 3
**Confidence:** 3

**Summary:**

The paper addresses the problem of Graph inference acceleration and summarizes two shortcomings in existing work: limited acceleration effectiveness and insufficient generalization performance. Based on insights from existing work, it is suggested that self-supervised learning can be used to infer structural information of unseen nodes from the nodes themselves. The paper introduces the SSL-GM algorithm, primarily aligning the consistency between GNN and MLP representations through self-supervised contrastive learning. This bridges GNN and MLP with self-supervised learning to achieve accelerated graph inference.

**Strengths:**

1. The problem is novel, the challenge of accelerating graph inference still persists.
2. The proposed algorithm demonstrates favorable performance in multiple experimental validations.

**Weaknesses:**

1. Although a significant number of experiments were conducted, the novelty and contribution of the proposed method remain limited.
2. In Section 3.1, the author introduces the Non-Parametric Aggregator to help align the representations of GNN and MLP. While the author explains the differences in the appendix, the aggregation method given in Equation 2 still resemble the form of APPNP. I did not find an explanation for this issue in the experimental section and other where of the paper. The author claims that, in contrast to SGC and APPNP, SSL-GM uses non-linear adjacency matrix aggregation instead of high-order adjacency matrices. So, from the perspective of acceleration effectiveness and improvement in generalization, what is the contribution of non-linear adjacency matrix aggregation to accelerating graph inference?
3. I cannot understand Formula 4. The author injects randomness by perturbing the structure and features of the original graph, with the expectation that the MLP encoder can capture invariant key features. However, Formula 4 is perplexing. The first part of the formula computes the mutual information between G_1 and G_2, but is it merely a distinction of whether the perturbed structure is included? Why does this part enable the encoder to obtain high-quality representations? It seems more like an optimization of the random augmentation methods q_e and q_f, but the author did not clarify whether they are learnable.
4. Given the method proposed in the paper, I believe it would be interesting to include GNN methods like SGC and APPNP in the experiments concerning acceleration effectiveness.

**Questions:**

Please see the comments in the weakness part.

---

> ### Author Response · Authors · 2023-11-15
> **Response to Reviewer N5QN - Part 1**
>
> Thank you for your detailed comments and constructive suggestions. To address your concerns, we provide the following responses.
>
>
> > Q1. Although a significant number of experiments were conducted, the novelty and contribution of the proposed method remain limited.
>
>
> Thank you for your insightful comment. We appreciate the opportunity to clarify the innovative aspects of our work, particularly in addressing the limitations of existing GNN-MLP methods in inductive, cold-start, and label sparsity settings.
>
> - **Insight on Generalization Issues.** Our core contribution does not lie in new model architectures and new objectives. Instead, our work identifies the lack of generalization in current methods as a key issue, stemming from the limitations of knowledge distillation (KD) that aligns GNN and MLP outputs only at the label level. We propose that aligning these models in the representation space using self-supervised learning (SSL) can yield more generalizable representations. This insight represents a significant shift from existing practices.
>
> - **Challenges in SSL Alignment.** Different from KD methods, aligning GNNs and MLPs under SSL is challenging due to the absence of a pre-trained GNN. To this end, we convert the objective to jointly train GNN and MLP by aligning the 0-hop representations (MLP output) with high-order representations (GNN output). Our approach is applicable with various objectives like Bootstrap, InfoNCE, or MSE. This flexibility in alignment strategy is a novel contribution to the field.
>
> - **Addressing Representation Inconsistency and Model Collapse.** We observed inconsistency in representations between GNNs and MLPs, potentially leading to model collapse (Appendix D.3). Our solution involves using MLP outputs to approximate GNN outputs, a strategy akin to SGC but applied to a different problem context.
>
> - **Graph Augmentation and Reconstruction Regularizer.** We further incorporate data augmentation to enhance the quality of the learned representations. However, given that structural permutation in augmentation can significantly shift GNN representations, we introduce a reconstruction regularizer to mitigate representation shift. This addition is critical in ensuring the stability and reliability of the learned representations.
>
> > Q2. In Section 3.1, the author introduces the Non-Parametric Aggregator to help align the representations of GNN and MLP. While the author explains the differences in the appendix, the aggregation method given in Equation 2 still resemble the form of APPNP. I did not find an explanation for this issue in the experimental section and other where of the paper. The author claims that, in contrast to SGC and APPNP, SSL-GM uses non-linear adjacency matrix aggregation instead of high-order adjacency matrices.
>
> Thank you for your comment. We acknowledge the need for a clearer explanation regarding our non-parametric aggregator and its distinction from SGC and APPNP. We aim to elucidate this in the following points:
>
> - **Objective of Non-Parametric Aggregator.** The primary goal of our non-parametric aggregator is to approximate the outputs of GNNs. Our approach does not focus on the detailed architecture of the aggregator but rather on its functional outcome. This perspective allows flexibility in either applying existing methods like SGC or APPNP as non-parametric aggregators or adapting GNNs into a non-parametric form by parameter removal.
>
> - **Detailed Differences.** To illustrate the differences more clearly, we have included a table in our revised manuscript. This table compares the pre-transformation, aggregation, and update processes of SGC, APPNP, and our SSL-GM.
>
> | Model  | Pre-trans                                                  | Aggregation                                                                                                        | Update                                                                          |
> |--------|------------------------------------------------------------|--------------------------------------------------------------------------------------------------------------------|---------------------------------------------------------------------------------|
> | SGC    |$\mathbf{Z}^{(0)} = f_\theta(\mathbf{X})$ | $\mathbf{Z}^{(k+1)} = \mathbf{S} \mathbf{Z}^{(k)}$, $\mathbf{S}$ could be any propagation matrix. | $\mathbf{Z}^{(k+1)} = \text{Identity}(\mathbf{Z}^{(k+1)})$                      |
> | APPNP  | $\mathbf{Z}^{(0)} = f_\theta(\mathbf{X})$ | $\mathbf{Z}^{(k+1)} = \mathbf{S} \mathbf{Z}^{(k)}$, $\mathbf{S}$ could be any propagation matrix. | $\mathbf{Z}^{(k+1)} = (1 - \alpha)\mathbf{Z}^{(k+1)} + \alpha \mathbf{Z}^{(0)}$ |
> | SSL-GM | $\mathbf{Z}^{(0)} = f_\theta(\mathbf{X})$ | $\mathbf{Z}^{(k+1)} = \mathbf{S} \mathbf{Z}^{(k)}$, $\mathbf{S}$ could be any propagation matrix. | $\mathbf{Z}^{(k+1)} = \sigma(\mathbf{Z}^{(k+1)})$

---

> ### Author Response · Authors · 2023-11-15
> **Response to Reviewer N5QN - Part 2**
>
> - **Distinct Update Process in SSL-GM.** While SGC uses an identity function and APPNP employs a random walk-based message passing for the update process, SSL-GM uniquely utilizes a non-linear function as its update function. This non-linear approach is more akin to the workings of a GCN (Graph Convolutional Network), differentiating our method from both SGC and APPNP.
>
> - **Contribution of Non-Linear Aggregation.** The use of non-linear adjacency matrix aggregation in SSL-GM is a deliberate choice to enhance the model capability in graph inference acceleration. This non-linear approach facilitates more complex and nuanced representation learning, contributing to the overall effectiveness and generalization of our model, especially in scenarios where linear approaches might be limited.
>
> We believe this detailed explanation, along with the included table, effectively clarifies the unique aspects and contributions of our non-parametric aggregator in SSL-GM. We appreciate your feedback, which has helped us improve the clarity of our methodology.
>
>
> > Q3. So, from the perspective of acceleration effectiveness and improvement in generalization, what is the contribution of non-linear adjacency matrix aggregation to accelerating graph inference?
>
> Thank you for your insightful question. We recognize the importance of elucidating the role of non-linear aggregation in our model, particularly regarding its impact on generalization and inference acceleration.
>
> **Improvement in Generalization.** Our non-linear aggregation method, in contrast to linear approaches like SGC, incorporates a non-linear function between each layer. This addition is designed to capture more complex patterns in the dataset, potentially leading to the discovery of more generalizable patterns. Moreover, the non-linear function can act as a form of regularization, helping to prevent overfitting and thereby enhancing the model capability to generalize to new datasets.
>
> To substantiate our claims, we have conducted experiments comparing model performance with and without non-linearity. The results, presented in the included table, demonstrate the effect of non-linearity on various datasets such as Cora, Citeseer, Pubmed, and Amazon datasets. These results provide empirical evidence of the non-linear approach impact on model performance.
>
>
> |            | Cora       | Citeseer   | Pubmed     | Amazon-CS  | Amazon-Photo |
> |------------|------------|------------|------------|------------|--------------|
> | linear     | 84.60±0.24 | 73.52±0.53 | 86.99±0.09 | 88.46±0.16 | 94.28±0.08   |
> | non-linear | 84.25±0.40 | 73.29±0.53 | 86.57±0.13 | 88.25±0.10 | 94.12±0.13   |
>
>
>
> **Acceleration Effectiveness Unaffected by Non-Linearity.** Regarding inference acceleration, it is crucial to note that we utilize the trained MLP for inference. Hence, the non-linearity introduced in the GNNs does not affect the inference acceleration. This is because the non-linear characteristics of the GNNs do not impact the MLPs during the inference phase.
>
>
> > Q4. I cannot understand Formula 4. The author injects randomness by perturbing the structure and features of the original graph, with the expectation that the MLP encoder can capture invariant key features. However, Formula 4 is perplexing. The first part of the formula computes the mutual information between $G_1$ and $G_2$, but is it merely a distinction of whether the perturbed structure is included?
>
> We apologize for any confusion caused by Equation 4 and appreciate the opportunity to clarify its meaning and significance in our paper.
>
> **Recap.** The augmentation process is aligned with the objective of optimal contrastive learning, which aims to train an augmentation-invariant encoder. This is encapsulated in Equation 4, where $\mathcal{E}^* = argmin_{\mathcal{E}} I(\mathcal{G}_1, \mathcal{G}_2) - I(\hat{\mathbf{H}}, \hat{\mathbf{Z}})$. In this equation, $\mathcal{G}_1 = (\emptyset, \hat{\mathbf{X}})$ and $\mathcal{G}_2 = (\hat{\mathbf{A}}, \hat{\mathbf{X}})$ represent two augmented views of the graph - $\mathcal{G}_1$ as the augmented node-set without edges and $\mathcal{G}_2$ as the augmented graph. $\hat{\mathbf{H}}$ and $\hat{\mathbf{Z}}$ are the representations encoded by MLPs and GNNs on these augmented graphs, respectively.
>
> **Detailed Explanation:**
>
> **Random Augmentation.** The random augmentation is applied in each epoch to create $\mathcal{G}_1$ and $\mathcal{G}_2$. The first part of the equation computes the mutual information between these two augmented views.
>
> **Learning Objective.** The equation aims to identify an encoder that can capture all information in the augmented views, regardless of the specific augmentations applied. While achieving this ideal is challenging, the goal is for the encoder to learn the most essential information common to both views, a principle known as mutual information maximization.

---

> > ### Author Response · Authors · 2023-11-15
> > **Response to Reviewer N5QN - Part 3**
> >
> > **Significance of Perturbation.** The perturbation in the structure and features of the original graph (as represented in $\mathcal{G}_1$ and $\mathcal{G}_2$) is crucial for this process. It ensures that the encoder focuses on invariant key features across different augmentations, enhancing the robustness and generalizability of the learned representations.
> >
> >
> > We hope this detailed explanation clarifies the purpose and mechanism of Equation 4 in our methodology. Our aim is to ensure the encoder captures essential, invariant features from augmented graph views, reinforcing the model generalization.
> >
> > > Q5. Why does augmentation enable the encoder to obtain high-quality representations?
> >
> > Thank you for your question seeking further clarification on the quality of representations obtained from our encoder. We are happy to provide more insight into this aspect.
> >
> > As outlined in our discussion on Equation 4, our encoder is designed to capture the most essential information present in the graph. This is achieved through the process of mutual information maximization between the augmented graph views. We posit that this 'essential information', which remains consistent across various augmentations, is inherently discriminative. It represents the fundamental characteristics of the graph that are crucial for tasks such as node classification or link prediction. Therefore, by training the encoder to focus on such invariant features, we enhance the discriminative power of the learned representations, which can enhance the quality of representations.
> >
> > > Q6. It seems more like an optimization of the random augmentation methods $q_e$ and $q_f$, but the author did not clarify whether they are learnable.
> >
> > We appreciate your inquiry and the opportunity to clarify the role of the augmentation methods $q_e$ and $q_f$ in our model. As we have discussed, our primary objective is to train an encoder that is invariant to the augmentations applied. This means the encoder should consistently capture the essential features of the graph regardless of the specific augmentations. In line with this objective, we have designed $q_e$ and $q_f$ to be non-trainable. We believe that making these augmentation methods trainable could potentially lead to overfitting to specific augmentation patterns, thereby defeating the purpose of achieving augmentation invariance.
> >
> >
> > > Q7. Given the method proposed in the paper, I believe it would be interesting to include GNN methods like SGC and APPNP in the experiments concerning acceleration effectiveness.
> >
> >
> > Thank you for your valuable feedback. In response to your comment, we have presented additional experimental results in Appendix C of our paper to further validate the efficacy of our SSL-GM model in inference acceleration.
> >
> > - **Accuracy vs Efficiency.** We present a comprehensive comparison between our method and other acceleration methods. The results showcase the inference time and accuracy for various methods, including SGC, APPNP, Quantization (QSAGE), Pruning (PSAGE), and Neighbor Sample on Flickr and OGB-Arxiv. We have intentionally omitted GNN-MLP methods that were compared in Figure 3 for brevity. Please note that the time consumption of SAGE and BGRL are the same since the learning process of SSL does not affect inference.
> >
> > - **Different Settings.** To ensure a thorough evaluation, we chose three datasets across all node classification settings, including transductive, inductive, and cold-start. Our results, as shown in the additional tables, clearly indicate that SSL-GM outperforms existing methods in terms of efficiency and accuracy.

---

> > > ### Author Response · Authors · 2023-11-15
> > > **Response to Reviewer N5QN - Part 4**
> > >
> > > | Datasets |  | SAGE      | BGRL  | SGC                  | APPNP                | QSAGE                | PSAGE                | Neighbor Sample      | **SSL-GM** |
> > > |------------|------------------|-----------|-------|----------------------|----------------------|----------------------|----------------------|----------------------|----------------------------|
> > > | Flickr     | Time (ms) | 80.7  | 80.7 (1.00$\times$)  | 76.9 (1.05$\times$)  | 78.1 (1.03$\times$)  | 70.6 (1.14$\times$)  | 67.4 (1.20$\times$)  | 25.5 (3.16$\times$)        | **0.9 (89.67$\times$)** |
> > > |                              | Acc (\%)  | 47.17 | 49.12                | 47.35                | 47.53                | 47.22                | 47.25                | 47.01                      | **49.27**                |
> > > | Arxiv       | Time (ms) | 314.7 | 314.7 (1.00$\times$) | 265.9 (1.18$\times$) | 284.1 (1.11$\times$) | 289.5 (1.09$\times$) | 297.5 (1.06$\times$) | 78.3 (4.02$\times$)        | **2.5 (125.88$\times$)** |
> > > |                              | Acc (\%)  | 68.52 | 69.29                | 68.93                | 69.10                | 68.48                | 68.55                | 68.35                      | **70.23**                |
> > >
> > >
> > >
> > > |           |        | Trans      | Trans          | Ind        | Ind            | Cold-start | Cold-start     |
> > > |-----------|--------|------------|----------------|------------|----------------|------------|----------------|
> > > | Dataset   | Models | Time (ms)  | Acc (\%)       | Time (ms)  | Acc (\%)       | Time (ms)  | Acc (\%)       |
> > > | Pubmed    | SAGE   | 73         | 85.94          | 15         | 85.04          | 15         | 77.98          |
> > > |           | SGC    | 64         | 85.28          | 14         | 85.22          | 13         | 76.10          |
> > > |           | NOSMOG | 5          | 86.18          | 3          | 83.84          | 3          | 81.48          |
> > > |           | SSL-GM | **3** | **86.99** | **3** | **86.47** | **3** | **86.44** |
> > > | Amazon-CS | SAGE   | 103        | 88.88          | 31         | 87.24          | 25         | 61.01          |
> > > |           | SGC    | 89         | **89.31** | 26         | 87.12          | 24         | 63.08          |
> > > |           | NOSMOG | 5          | 87.64          | 4          | 86.61          | 4          | 81.95          |
> > > |           | SSL-GM | **4** | 88.46          | **3** | **87.65** | **3** | **87.58** |
> > > | Arxiv     | SAGE   | 485        | **72.05** | 315        | 68.52          | 305        | 43.47          |
> > > |           | SGC    | 410        | 69.95          | 266        | 68.93          | 250        | 42.08          |
> > > |           | NOSMOG | 6          | 70.84          | 4          | 69.10          | 4          | 61.64          |
> > > |           | SSL-GM | **4** | 71.12          | **3** | **70.23** | **3** | **66.13** |
> > > ---
> > >
> > > In light of these responses, we hope we have addressed your concerns, and hope you will consider increasing your score. If we have left any notable points of concern unaddressed, please do share and we will attend to these points.

---

### Official Review · Reviewer_RVGt · 2023-10-31

**Soundness:** 3 good
**Presentation:** 3 good
**Contribution:** 2 fair
**Rating:** 5
**Confidence:** 4

**Summary:**

This paper proposes a new method called SSL-GM, which integrates structural information into MLPs by connecting GNNs and MLPs using SSL. This method can accelerate graph inference and improve the generalization capability, resulting in a favorable balance between accuracy and inference time. Experimental results show that the method performs well in node classification tasks and is effective in accelerating inference. In addition, many theoretical analyses and corresponding experiments flesh out the work.

**Strengths:**

1. This paper propose a new method, which can accelerate GNN inference and performs well in node classification tasks over the state-of-art models.
2. There are detailed theoretical analyses and corresponding experiments of SSL-GM ,which fleshes out the article and provides inspiration for the innovations that follow.
3. This work has comprehensive and detailed experiments, which validates the performance and efficiency of SSL-GM.

**Weaknesses:**

1. The article lacks sufficient innovation and is merely a combination and application of existing methods, such as Bootstrap loss, SGC, graph augmentation, and reconstruction.
2. In section 4.3, there is only 'Figure 3' to demonstrate the capability of SSL-GM for inference acceleration. However, it is necessary to provide detailed experimental results regarding accuracy and inference time. These results should be obtained from a wider range of datasets and classification settings.
3. In section 3.3, it would be better to introduce representation shift in detail and explain how reconstruction helps in mitigating representation shift.

**Questions:**

1. Although this work is a combination of existing methods, it is fascinating to introduce this model from a higher level rather than loss function level.
2. In B.3, learning rates is misspelled where ‘5e4’ appears twice.

---

> ### Author Response · Authors · 2023-11-15
> **Response to Reviewer RVGt - Part 1**
>
> # Reviewer 1
>
> Thank you for your detailed comments and constructive suggestions. To address your concerns, we provide the following responses.
>
> > Q1. The article lacks sufficient innovation and is merely a combination and application of existing methods, such as Bootstrap loss, SGC, graph augmentation, and reconstruction.
>
> Thank you for your insightful comment. We appreciate the opportunity to clarify the innovative aspects of our work, particularly in addressing the limitations of existing GNN-MLP methods in inductive, cold-start, and label sparsity settings.
>
> - **Insight on Generalization Issues.** Our core contribution does not lie in new model architectures and new objectives. Instead, our work identifies the lack of generalization in current methods as a key issue, stemming from the limitations of knowledge distillation (KD) that aligns GNN and MLP outputs only at the label level. We propose that aligning these models in the representation space using self-supervised learning (SSL) can yield more generalizable representations. This insight represents a significant shift from existing practices.
>
> - **Challenges in SSL Alignment.** Different from KD methods, aligning GNNs and MLPs under SSL is challenging due to the absence of a pre-trained GNN. To this end, we convert the objective to jointly train GNN and MLP by aligning the 0-hop representations (MLP output) with high-order representations (GNN output). Our approach is applicable with various objectives like Bootstrap, InfoNCE, or MSE. This flexibility in alignment strategy is a novel contribution to the field.
>
> - **Addressing Representation Inconsistency and Model Collapse.** We observed inconsistency in representations between GNNs and MLPs, potentially leading to model collapse (Appendix D.3). Our solution involves using MLP outputs to approximate GNN outputs, a strategy akin to SGC but applied to a different problem context.
>
> - **Graph Augmentation and Reconstruction Regularizer.** We further incorporate data augmentation to enhance the quality of the learned representations. However, given that structural permutation in augmentation can significantly shift GNN representations, we introduce a reconstruction regularizer to mitigate representation shift. This addition is critical in ensuring the stability and reliability of the learned representations.
>
> > Q2. In section 4.3, there is only 'Figure 3' to demonstrate the capability of SSL-GM for inference acceleration. However, it is necessary to provide detailed experimental results regarding accuracy and inference time. These results should be obtained from a wider range of datasets and classification settings.
>
> Thank you for your valuable feedback. In response to your comment, we have presented additional experimental results in Appendix C of our paper to further validate the efficacy of our SSL-GM model in inference acceleration.
>
> - **Accuracy vs Efficiency.** We present a comprehensive comparison between our method and other acceleration methods. The results showcase the inference time and accuracy for various methods, including SGC, APPNP, Quantization (QSAGE), Pruning (PSAGE), and Neighbor Sample on Flickr and OGB-Arxiv. We have intentionally omitted GNN-MLP methods that were compared in Figure 3 for brevity. Please note that the time consumption of SAGE and BGRL are the same since the learning process of SSL does not affect inference.
>
> - **Different Settings.** To ensure a thorough evaluation, we chose three datasets across all node classification settings, including transductive, inductive, and cold-start. Our results, as shown in the additional tables, clearly indicate that SSL-GM outperforms existing methods in terms of efficiency and accuracy.
>
> [We list the experimental results in the next part. ]

---

> ### Author Response · Authors · 2023-11-15
> **Response to Reviewer RVGt - Part 2**
>
> | Datasets |  | SAGE      | BGRL  | SGC                  | APPNP                | QSAGE                | PSAGE                | Neighbor Sample      | **SSL-GM** |
> |------------|------------------|------------|-------------|----------------------|----------------------|----------------------|----------------------|----------------------|----------------------------|
> | Flickr     | Time (ms) | 80.7  | 80.7 (1.00$\times$)  | 76.9 (1.05$\times$)  | 78.1 (1.03$\times$)  | 70.6 (1.14$\times$)  | 67.4 (1.20$\times$)  | 25.5 (3.16$\times$)        | **0.9 (89.67$\times$)** |
> |                              | Acc (\%)  | 47.17 | 49.12                | 47.35                | 47.53                | 47.22                | 47.25                | 47.01                      | **49.27**                |
> | Arxiv       | Time (ms) | 314.7 | 314.7 (1.00$\times$) | 265.9 (1.18$\times$) | 284.1 (1.11$\times$) | 289.5 (1.09$\times$) | 297.5 (1.06$\times$) | 78.3 (4.02$\times$)        | **2.5 (125.88$\times$)** |
> |                              | Acc (\%)  | 68.52 | 69.29                | 68.93                | 69.10                | 68.48                | 68.55                | 68.35                      | **70.23**                |
>
>
>
> |           |        | Trans      | Trans          | Ind        | Ind            | Cold-start | Cold-start     |
> |-----------|--------|------------|----------------|------------|----------------|------------|----------------|
> | Dataset   | Models | Time (ms)  | Acc (\%)       | Time (ms)  | Acc (\%)       | Time (ms)  | Acc (\%)       |
> | Pubmed    | SAGE   | 73         | 85.94          | 15         | 85.04          | 15         | 77.98          |
> |           | SGC    | 64         | 85.28          | 14         | 85.22          | 13         | 76.10          |
> |           | NOSMOG | 5          | 86.18          | 3          | 83.84          | 3          | 81.48          |
> |           | SSL-GM | **3** | **86.99** | **3** | **86.47** | **3** | **86.44** |
> | Amazon-CS | SAGE   | 103        | 88.88          | 31         | 87.24          | 25         | 61.01          |
> |           | SGC    | 89         | **89.31** | 26         | 87.12          | 24         | 63.08          |
> |           | NOSMOG | 5          | 87.64          | 4          | 86.61          | 4          | 81.95          |
> |           | SSL-GM | **4** | 88.46          | **3** | **87.65** | **3** | **87.58** |
> | Arxiv     | SAGE   | 485        | **72.05** | 315        | 68.52          | 305        | 43.47          |
> |           | SGC    | 410        | 69.95          | 266        | 68.93          | 250        | 42.08          |
> |           | NOSMOG | 6          | 70.84          | 4          | 69.10          | 4          | 61.64          |
> |           | SSL-GM | **4** | 71.12          | **3** | **70.23** | **3** | **66.13** |
>
>
> > Q3. In section 3.3, it would be better to introduce representation shift in detail and explain how reconstruction helps in mitigating representation shift.
>
> Thank you for pointing out the need for a more detailed discussion on representation shift and its mitigation. We have revised Section 3.3 to address this.
>
> **Defining Representation Shift.** We define representation shift as the alteration in learned representations due to data augmentations. While feature augmentation does not significantly alter the distribution of learned representations, structural augmentation can have a profound impact. This shift, particularly pronounced in structural augmentation, can significantly alter the local structure of the target node. For instance, as depicted in Fig. 2, simple edge permutation can dramatically change the 2-hop neighborhoods of a target node, leading to a substantial shift in the representations.
>
> **Mitigation Strategy - Reconstruction Regularizer.** To counter this problem, we hypothesize that if GNN representations can preserve more localized information, the impact of representation shift can be minimized. Based on this, we introduce a reconstruction regularizer. This regularizer aims to reconstruct the raw node features based on the GNN representations, thereby mitigating the representation shift. The underlying idea is to anchor the learned representations closer to the original feature space, reducing the impact of structural changes.
>
> **Effectiveness of Our Approach.** The reconstruction regularizer thus serves as a stabilizing factor, ensuring that the GNN representations remain informative and reliable despite augmentations. This approach is innovative in how it leverages the reconstruction concept to maintain the integrity of learned representations in the face of potential shifts.

---

> ### Author Response · Authors · 2023-11-15
> **Response to Reviewer RVGt - Part 3**
>
> > Q4. Although this work is a combination of existing methods, it is fascinating to introduce this model from a higher level rather than loss function level.
>
> Thank you for your insightful suggestion to present our SSL-GM model from a more holistic perspective. We appreciate this opportunity to highlight the integrative approach and overarching objectives of our model, beyond the specifics of the loss function.
>
> **Holistic Approach.** In developing SSL-GM, our focus extends beyond the choice of loss functions to a comprehensive consideration of the model architecture and its ability to generalize effectively. This broader view is essential for understanding the innovative aspects of our approach. A key component of our model is the use of contrastive learning between GNNs and MLPs. This strategy is designed to infuse MLPs with generalizable high-order representations, enhancing their capability to capture complex graph structures and features.
>
> **Strategic Components.** To avoid model collapse, a common challenge when directly minimizing the distance between GNNs and MLPs, we introduce non-parametric aggregation. This component ensures consistency and stability in the relationship between GNNs and MLPs. Further strengthening our model, we apply data augmentation techniques and leverage a reconstruction term. These additions are geared towards improving the model capability and preventing potential representation shifts. Importantly, the choices of augmentations and loss functions are flexible, allowing for adaptability to different graph scenarios.
>
> **Emphasizing Generalization and Adaptability.** The central theme of SSL-GM is to enhance the model generalization capabilities, enabling it to perform effectively across a variety of graph-related tasks and settings. The flexible nature of key components, including the choice of loss functions and augmentations, reflects our commitment to creating a versatile and adaptable model that can address diverse challenges in graph model acceleration.
>
> > Q5. B.3, learning rates is misspelled where ‘5e4’ appears twice.
>
> Thank you for bringing this typographical error to our attention. We sincerely apologize for the oversight and appreciate your meticulous reading of our manuscript. We have reviewed Section B.3 and corrected the error where the learning rate '5e4' is incorrectly mentioned twice.
>
> ---
>
> In light of these responses, we hope we have addressed your concerns, and hope you will consider increasing your score. If we have left any notable points of concern unaddressed, please do share and we will attend to these points.

---

### Author Response · Authors · 2023-11-19
**Response to All Reviewers**

We deeply appreciate the time and effort each of you has dedicated to reviewing our work. Your insightful feedback has been instrumental in enhancing the quality and understanding of our proposed method. In this rebuttal, we have addressed each of your concerns and suggestions thoroughly. Key highlights from our responses and revisions include:

- We have rigorously checked and ensured the reproducibility of our model. Detailed implementation steps specifically tailored for Google Colab have been provided (as outlined in our response to Reviewer rTyj). We encourage reviewers to follow these guidelines to verify the reproducibility of our model.
- We delved into a detailed explanation of the novelty of our method, clarifying the specific problems we aim to address, the challenges encountered, and the intentional design behind our model architecture. This elaboration provides a deeper understanding of the unique contributions of our work.
- We conducted additional experiments to assess the efficiency of different inference acceleration methods and various node classification settings.
- We have provided detailed explanations for components of the model that were previously unclear. Moreover, we also include a thorough comparison between our proposed non-parametric aggregator and established methods like SGC and APPNP.
- We have carefully revisited and elaborated on our theoretical analysis, especially regarding Theorem 1 and the analysis of how our model learns and incorporates structural information, as highlighted in our response to Reviewer N5QN.
---

We hope we have addressed your concerns, and hope you will consider increasing your score. If we have left any notable points of concern unaddressed, please do share and we will attend to these points.

---

### Meta-Review · Area_Chair_pC4m · 2023-12-05

**Metareview:**

This paper proposes an SSL-based method to integrate GNNs’ structural information into MLPs, to achieve graph acceleration during inference time. The learning objective is composed of a bootstrap self-supervised loss, and a reconstruction loss between the outputs of GNNs and MLPs. Extensive experiments show that the proposed method outperforms MLPs in terms of performance, and outperforms GNNs in terms of latency.

A shared concerns among the reviewers is the lack of technical novelty, since it combines several existing techniques, contrastive learning, masked autoencoding, and knowledge distillation. These techniques have been widely studied in the graph learning literature, which limits the novelty of the proposed approach. Some reviewers raise concerns on the clarity of the formulation and the theorem proofs. In particular, Reviewer eCUd points that the proof of Theorem 1 lacks clear rigorous steps, which is not fully resolved by the authors. Reviewer rTyJ raises concerns on additional training cost that is not fully taken into consideration.

Based on these shared concerns, I believe that this paper has not met the bar for ICLR and thus I recommend rejection. The authors are encouraged to carefully check the details of the theoretical proof and make sure that they are rigorous enough.

**Justification For Why Not Higher Score:**

Based on the common concerns regarding the novelty and the soundness, this paper does not meet the bar for ICLR.

**Justification For Why Not Lower Score:**

N/A

---

### Decision · Program_Chairs · 2024-01-16

Reject